# DP-RFT: Learning to Generate Synthetic Text via Differentially Private Reinforcement Fine-Tuning

## Abstract

Differentially private (DP) synthetic data generation plays a pivotal role in advancing applications of large language model (LLM) involving private user data. Generating synthetic data that resemble *eyes-off* private corpora involves a difficult trade-off. On one hand, DP finetuning methods offers formal privacy guarantees, yet requires the data owner to expose the raw content for model training. However, methods that avoid direct exposure to private data are bounded by an off-the-shelf, un-finetuned model, whose outputs often lack domain fidelity. **Can we *train* an LLM to generate high-quality synthetic text without providing it individual private examples?** In this work, we introduce *Differentially Private Reinforcement Fine-Tuning* (DP-RFT), an online reinforcement learning algorithm for synthetic data generation with LLMs. DP-RFT leverages DP-protected nearest-neighbor votes from an eyes-off private corpus as a reward signal for on-policy synthetic samples generated by an LLM. The LLM iteratively learns to generate synthetic data to maximize the expected DP votes through Proximal Policy Optimization (PPO). We evaluate DP-RFT for long-form and domain-specific synthetic data generation, such as news articles, meeting transcripts, and medical article abstracts. Our experiments show that DP-RFT significantly outperforms private evolution and DP fine-tuning methods, such as DP-SGD, in terms of both the fidelity and downstream utility of the generated synthetic data.

## 1 Introduction

Large language models (LLMs) have achieved remarkable progress in their capabilities. At the core of this progress lies *data*: the scale and quality of training and evaluation data heavily impact the effectiveness of these models (Kaplan et al., 2020; Biderman et al., 2023b). However, as LLMs are increasingly trained on real-world data, there are growing concerns around the responsible use of customer or copyrighted data (Carlini et al., 2024; Panda et al., 2025). LLMs trained on private or sensitive data can memorize these patterns, causing security and privacy risks by exposing private information through the model outputs (Biderman et al., 2023a; Chen et al., 2024; Shi et al., 2024).

*Differentially-private* (DP) synthetic data has emerged as a promising direction for responsible data use (Jordon et al., 2018; Li et al., 2022; Yu et al., 2022; Yue et al., 2023; Harder et al., 2023; Lin et al., 2024; Xie et al., 2024). The goal is to leverage a language model to generate a synthetic dataset that is statistically similar to the private dataset, while ensuring that individual samples in the original data cannot be inferred with high confidence from the model outputs (Dwork et al., 2006). DP synthetic data can be used for downstream model development (e.g. training, evaluation, etc.) with intact privacy guarantee thanks to DP's post-processing properties (Dwork et al., 2014).

Using LLM to generate DP synthetic text often involves a difficult trade-off. On one hand, existing methods that rely on training or finetuning against private dataset, e.g. DP-SGD (Abadi et al., 2016), usually require direct access to raw private data during training, which may be infeasible under "eyes-off" constraints settings where data custodians cannot expose or share individual records (Ponomareva et al., 2023). On the other hand, existing methods that avoid direct exposure to private data, e.g. AugPE (Xie et al., 2024), are bounded by samples generated from an off-the-shelf, frozen LLM, whose outputs often lack domain fidelity.

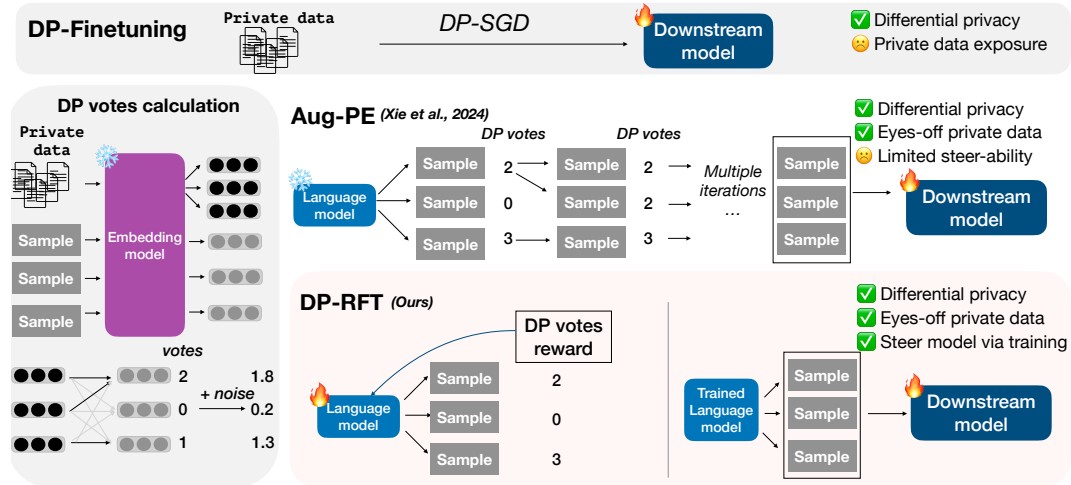

Figure 1: An illustration of DP-RFT and comparison with prior methods. DP-RFT fine-tunes a LM to generate texts similar to the private document with reinforcement learning, guided by a DP-protected nearest-neighbor votes as reward. Compared to baseline methods (DP-finetuning and Aug-PE), DP-RFT generates synthetic data with higher fidelity by training an LM to generate texts more similar to the private data while preserving the *eyes-off* requirement of private data.

To get the best of both worlds, we propose *Differentially Private Reinforcement Fine-Tuning* (DP-RFT), a reinforcement learning (RL) algorithm for training LLMs for synthetic data generation where the LLM *never* ingests private examples directly. Building upon the reward-based RL post-training techniques for LLMs (Ouyang et al., 2022; Zelikman et al., 2022; Jaech et al., 2024; Guo et al., 2025), DP-RFT leverages DP-protected nearest-neighbor votes (Lin et al., 2024; Xie et al., 2024) from an eyes-off private corpus as a reward signal for LLM's on-policy synthetic sample generation. The LLM iteratively learns from reward feedback to generate synthetic data to maximize the expected DP votes through Proximal Policy Optimization (PPO; Schulman et al., 2017).

We evaluate DP-RFT for *long-form* synthetic data generation with different domains and types of text, such as news articles (Narayan et al., 2018), meeting transcripts (Zhong et al., 2021), LLM-user chat logs (Zhao et al., 2024) and medical article abstracts (Yu et al., 2023). We train DP-RFT models under different levels of privacy budget, and evaluate the generated synthetic data by intrinsic quality (§ 5.2), as well as downstream utility when the synthetic data is used as training data for another language model (§ 5.1). Our experiments show that DP-RFT outperforms both Aug-PE and DP-finetuning methods in terms of the fidelity and downstream utility of the generated synthetic data, where the performance gap can in-part be explained by whether the type or domain of data is "in-distribution" to the base, un-finetuned LLMs' output distribution. Our key contributions are:

1. *New finetuning method for DP synthetic data generation*: We propose DP-RFT , which leverages DP-protected reward function and reinforcement learning to train a LLM against private corpus, without *ever* ingesting private examples directly during the training process.

2. *In-depth experiments and analyses*: We conduct comprehensive evaluation of our methods on four datasets, showing significant improvement on downstream utility, especially for a tight privacy budget and on dataset which are out-of-distribution for the backbone language model. Our qualitative analysis further reveals that DP-RFT is able to better capture lexical and structural similarity of long-form, structural outputs.

## 2 BACKGROUND AND RELATED WORK

**Differential Privacy with LLMs.** As LLMs are increasingly deployed in many real-world cases, preserving the privacy of training and evaluation data has been an active research area. One popular approach is to apply DP-SGD (Abadi et al., 2016) to train LLMs for classification (Yu et al., 2021; Li et al., 2021) or generation tasks (Wang et al., 2025a; Yu et al., 2021; Li et al., 2021; Yue et al., 2023;

Mattern et al., 2022; Kurakin et al., 2023; Ngong et al., 2024; Tan et al., 2025). Later work proposed training-free approaches, such as injecting calibrated DP noise into the token-by-token generation process of LLMs (Tang et al., 2023; Flemings et al., 2025; Duan et al., 2023), or aggregating multiple LLM outputs in a DP manner (Wu et al., 2023). While these methods provide theoretical DP guarantees, they all require exposing the private data as input to the LLM. In practice, there could be regulatory constraint which requires the private data to be completely *eyes-off*, preventing the adoption of such methods. Private Evolution (PE) (Lin et al., 2024; Xie et al., 2024; Lin et al., 2025; Wang et al., 2025b) is a newly emerged framework to address this challenge. It proposed an iterative prompting pipeline which steers the model generation to be more similar to the private data, as measured by embedding similarity. While these methods avoid private inputs, their reliance on un-finetuned models limits the synthetic data quality. Our DP-RFT aims to take the best of both worlds: adapting model weights to better fit private data without directly inputting it into the model. Recent work(Nguyen et al., 2025) proposes to generate synthetic data via gradient matching, which requires gradient computation on private data. Notably, Hou et al. (2025) recently proposed fine-tuning LLMs using embedding similarity as a reward with an offline reinforcement learning algorithm (DPO; Rafailov et al., 2023). However, their work focuses on federated learning, while ours considers the centralized setting.

**Reinforcement fine-tuning (RFT) for LLMs.** We briefly review how reinforcement learning is used to post-train large language models. Let a language model with parameters $\theta$ define a policy $\pi_\theta(\cdot \mid p)$ over token sequences given a prompt $p$. The goal of RFT is to update $\theta$ so that samples $d \sim \pi_\theta(\cdot \mid p)$ have higher task-specific utility, quantified by a scalar reward $R(d, p)$. In modern LLM post-training, $R$ can come from a learned reward model (e.g., RLHF, Ouyang et al., 2022), or from verifiable signals (e.g., passing tests in code/math), and training proceeds by maximizing the cumulative rewards. Practically, RFT alternates between: (i) sampling on-policy generations from $\pi_\theta$ for a batch of prompts; (ii) computing scalar rewards for each sample; and (iii) updating $\theta$ via a policy gradient method (e.g., PPO, Schulman et al., 2017; or GRPO, Shao et al., 2024) on those samples. Compared to supervised fine-tuning (SFT), where LLM ingests reference texts as input during training, RFT directly optimizes the task metric exposed by $R$ and is thus effective when high-quality labels are scarce or when a reward function $R$ can be defined. The fact that RFT does not ingest example outputs during training is the key to our DP-RFT algorithm's design.

## 3 DP-RFT: DIFFERENTIALLY-PRIVATE REINFORCEMENT FINETUNING

Given a set of private documents $D_{\text{priv}}$, our goal is to train a language model $M_{\text{gen}}$ that generates synthetic documents $D_{\text{syn}}$ similar to $D_{\text{priv}}$. $M_{\text{gen}}$ takes an input prompt $p$ which contains public information about the private corpus and produces a synthetic document $d$. In this section, we describe our method, DP-RFT, which leverages reinforcement fine-tuning (RFT) to train $M_{\text{gen}}$ against a DP-protected reward function $R$.

### 3.1 GENERATING INPUT PROMPTS

We leverage a large language model (LLM) to construct input prompts which contain public information about the target domain. We construct prompts to encourage diverse generation from $M_{\text{gen}}$ following Augmented Private Evolution (Aug-PE) (Xie et al., 2024) for training and inference of DP-RFT. We focus on one or both of the two axes of diversity (1) content diversity and (2) length diversity. For content diversity, we generate a set of diverse keywords using an LLM. For instance, for BBC articles, we prompt an LLM to generate a list of topics (e.g. Politics, Sports), along with keywords associated with each topic. We obtain a set of prompts, each with a different set of keywords, which we use as input to $M_{gen}$. For length diversity, we define a possible range of length for the private documents and random sample one length to include in the prompt (e.g. "The generated document should contain around 300 words"). We include the details of the prompts and the procedure in §A.3 in the appendix.

### 3.2 REWARDS

**DP voting as reward.** Aug-PE (Xie et al., 2024) prompts language models to generate multiple synthetic documents and iteratively generate synthetic documents via a process called **DP voting**:

each private document votes for a synthetic document that is most similar to them measured with a similarity function, such as a text embedding model. They find that selecting synthetic documents that receive more votes (i.e. close to more private documents in the embedding space) lead to better synthetic documents and hence downstream performance. We take inspiration from Aug-PE and propose the following new designs to the DP-voting mechanism and the iterative generation-sampling process: **(1) Using raw similarity scores as votes.** In Aug-PE, each private sample votes only for its closest synthetic document, producing a binary (1 or 0) signal. This discards valuable information in the raw similarity scores, such as *how much* one synthetic sample is better than another. To capture this richer signal, we instead use the raw similarity scores $sim(d, D_{priv,i})$ directly as votes from the $i$-th private document $D_{priv,i}$ to the synthetic document $d$, , where $sim$ is a similarity measure (e.g. text embedding similarity). **(2) DP-RFT.** While Aug-PE can promote synthetic documents that are similar to the private documents, the model is untrained, and hence has limited steerability. We instead leverage the votes as the reward in Reinforcement Fine-tuning (RFT) to steer $M_{\text{gen}}$ to generate synthetic documents more similar to the private corpus. Concretely, for a synthetic document $d$ and a predefined similarity measure $sim$, we define $R_{sim} = \frac{1}{|D_{priv}|} \sum_{i=0}^{i=|D_{priv}|} sim(d, D_{priv,i})$ as the reward.

**Ensuring differential privacy.** To ensure differential privacy, we add Gaussian noise to $R_{sim}$ following Aug-PE. For a synthetic example $d$, we first obtain its similarity to each of the private documents $sim(d, D_{priv,i})$ and clip the similarity score to a threshold $c$. Given a noise multiplier $\sigma$ and a batch of synthetic samples of size $s$, we add Gaussian noise $\mathcal{N}(0, \sigma c \sqrt{s})$ to $\sum_i sim(d, D_{priv,i})$, i.e., the sum of similarity between the synthetic document and all private documents. Finally, we divide the sum of the number of private documents to obtain the average similarity scores. We provide the pseudocode of this procedure in Figure 2. The privacy analysis follows the Private Evolution framework (Lin et al., 2024), which is reproduced in §A.4 for completeness.

**Input:** A set of $s$ synthetic documents $D_{syn}$, a similarity metric $sim$, a set of $p$ private documents $D_{priv}$, a noise multiplier $\sigma$, a raw similarity threshold $c$.
**Output:** Reward $R_{sim}$ for synthetic documents $D_{syn}$

```
 1: R_sim ← []
 2: for i ∈ {1, . . . , s} do
 3:     raw_similarity ← []
 4:     for j ∈ {1, . . . , p} do
 5:         S_{i,j} ← sim(D_{syn,i}, D_{priv,j})
 6:         if σ > 0 then
 7:             S_{i,j} = min(S_{i,j}, c)
 8:         end if
 9:         raw_similarity.append(S_{i,j})
10:     end for
11:     raw_similarity = sum(raw_similarity)
12:     if σ > 0 then
13:         raw_similarity += N(0, σc√s)
14:     end if
15:     R_sim.append(raw_similarity / p)
16: end for
    Return: R_sim
```

Figure 2: Procedure for computing similarity reward ($R_{sim}$) given a noise multiplier $\sigma$ for a batch of $s$ synthetic samples.

**Mitigating reward hacking.** Although optimizing for $R_{sim}$ can lead to synthetic documents similar to the private corpus defined by $sim$, relying only on $R_{sim}$ as the sole reward can lead to reward hacking (Amodei et al., 2016; Pan et al., 2022). Unlike prior RFT methods which focus on math and coding problems (DeepSeek-AI, 2025; Wei et al., 2025) where a verifiable reward (e.g. presence of the gold answer string) can be defined, generating documents that are similar to a set of documents is relatively open-ended and unconstrained. Moreover, text embedding models are known to have various biases (Coelho et al., 2024; Fayyaz et al., 2025), especially when embedding *long* documents. Thus, in addition to $R_{sim}$, we employ a reward that measures adherence to the input prompt to prevent reward hacking.[1] We implement $R_{prompt}$ using a combination of LLM-as-a-judge and rule-based checks and describe the details in §4.

To combine the two rewards, we use a threshold approach. We set the reward to 0 if $R_{prompt}$ is less than a threshold $\tau$ and otherwise return $R_{sim}$. The final reward is defined as:

$$R = \begin{cases} R_{\text{sim}} & \text{if } R_{\text{prompt}} > \tau \\ 0 & \text{otherwise} \end{cases}$$

---

[1] While RL algorithms such as PPO includes a KL penalty term to reduce divergence from the reference model, we find that increasing the KL coefficient does not solve the issue. In preliminary experiments, we observe that it would discourage the model from learning to generate samples closer to the private distribution.

We train $M_{\text{gen}}$ with Proximal Policy Optimization (PPO, Schulman et al. (2017)).

## 4 EXPERIMENTS

### 4.1 EVALUATION SETTINGS

**Datasets.** We evaluate DP-RFT and the baselines on four publically available datasets that are treated as private datasets, covering different domains and document structures:

- *Domain-specific document-level data:* **PubMed**, which contains abstracts of medical papers crawled by Yu et al. (2023) from 2023/08/01 to 2023/08/07; **BBC News article** from 2010 to 2017 released by Narayan et al. (2018)

- *Structured long-form data:* **Wildchat** (Zhao et al., 2024), which contains chat history between a user and an LLM; collected between 2023 to 2024 and **QMSum** (Zhong et al., 2021) which contains meeting transcripts.

The average number of tokens for PubMed, BBC article, Wildchat and QMSum are $361, 513, 2,799$ and $2,857$ respectively. We include details about these datasets in §A.5.1 in the appendix.

We evaluate the generated synthetic documents for both intrinsic and extrinsic evaluation.

**Intrinsic evaluation: similarity with real data.** To measure similarity between the synthetic data and private data, we report quantitative metrics including the average and max embedding similarity with the private corpus and the embedding distribution distance, i.e., the Fréchet Inception Distance (Heusel et al., 2017), following prior work (Xie et al., 2024). As it is non-trivial to measure similarity of long documents, we further employ LLM-as-a-judge to evaluate pairwise similarity of two synthetic documents against a private document.

**Extrinsic evaluation: downstream performance.** Aside from intrinsic similarity evaluation, the synthetic text should be helpful in downstream utiliby. To measure downstream performance, we fine-tune a language model via next token prediction on the synthetic text generated and report the next token prediction accuracy on the test set from the private data. Following previous work (Yu et al., 2023; Xie et al., 2024; Hou et al., 2025), we fine-tune and evaluate BERT$_{\text{small}}$ (Devlin et al., 2019) as a causal language model by modifying bidirection attention to causal attention.

### 4.2 DP-RFT AND BASELINE SETTINGS

We provide the implementation details for each component for DP-RFT as described in §3.

$M_{\text{gen}}$ and $R_{sim}$: For all datasets, we use Qwen-2.5-3B-Instruct (Qwen Team, 2024) as $M_{\text{gen}}$ and gte-Qwen-2-1.5B-Instruct (Li et al., 2023) to measure $R_{sim}$. For WildChat and QMSum, we also measure the distributions of word-level jaccard similarity and word counts of all speech turn in a synthetic vs. private example, and use their KL divergence as two additional terms in $R_{sim}$, in order to encourage $M_{gen}$ to learn the structural properties of private corpus. We conduct ablation studies varying the choice of embedding model, language model, as well as the reward terms in § 6.2.

$R_{prompt}$: We use gpt-4o (OpenAI, 2024) as the LLM-as-a-judge to evaluate how well the output adheres to the prompt. Concretely, given the input prompt $p$ and generated documents $d$, the model outputs a scalar rating from 1 to 10 on how well $d$ adheres to $p$. We include the exact prompt in §A.3. To evaluate adherence to the length instruction for the BBC article and PubMed, we employ a rule-based approach that checks the difference between the length of the generated documents and the specified length in the prompt. If the length difference is greater than a threshold $d$ or the LLM-as-a-judge output is less than a threshold $\tau$, we set the reward to 0. We set the threshold $\tau$ to 6 and $d$ as $\{50, 100\}$ for PubMed and BBC articles respectively.

**Input prompts.** Following prior work (Xie et al., 2024), we construct input prompts for each dataset by prompting an LM to generate keywords based on public knowledge. We design a multi-stage pipeline to ensure diversity of the prompts. For BBC articles, we first prompt the language model to generate a list of category of BBC articles. We then prompt the models to generate keywords

Table 1: Evaluation on downstream model performance using Qwen-2.5-3B-Instruct as the synthetic data generator. DP-RFT outperform all baseline methods when there is a privacy constraint ($\epsilon \neq \infty$). We report the result of using the full private corpus to train for DP-FT as well as using the same amount of generated data (2,000) to train.

| Dataset | Method | Data Type (Size) | BERT$_{Small}$ Next Token Accuracy ($\uparrow$) | | | |
|---|---|---|---|---|---|---|
| | | | $\epsilon = \infty$ | $\epsilon = 4$ | $\epsilon = 2$ | $\epsilon = 1$ |
| **PubMed** | DP-FT | Private (75,316) | 44.88 | 18.41 | 17.99 | 17.63 |
| | DP-FT | Private (2,000) | **20.79** | 9 | 8 | 6 |
| | QWEN | Synthetic (2,000) | 16.88 | 16.88 | 16.88 | **16.88** |
| | Aug-PE | Synthetic (2,000) | 13.96 | 14.41 | 14.16 | 13.71 |
| | DP-RFT | Synthetic (2,000) | 17.46 | **17.81** | **17.32** | 16.79 |
| **BBC** | DP-FT | Private (10,000) | 25.75 | 14.07 | 13.71 | 13.09 |
| | DP-FT | Private (2,000) | **17.97** | 9.29 | 7.96 | 6.90 |
| | QWEN | Synthetic (2,000) | 13.07 | 13.07 | **13.07** | 13.07 |
| | Aug-PE | Synthetic (2,000) | 11.63 | 10.78 | 10.03 | 10.00 |
| | DP-RFT | Synthetic (2,000) | 13.72 | **13.25** | 13.03 | **13.51** |
| **Wildchat** | DP-FT | Private (10,000) | 28.78 | 13.93 | 13.52 | 12.80 |
| | DP-FT | Private (2,000) | **19.62** | 8.28 | 7.37 | 5.30 |
| | QWEN | Synthetic (2,000) | 5.22 | 5.22 | 5.22 | 5.22 |
| | Aug-PE | Synthetic (2000) | 13.29 | 14.04 | 13.81 | 13.63 |
| | DP-RFT | Synthetic (2000) | 13.93 | **14.08** | **14.14** | **13.89** |
| **QMSum** | DP-FT | Private (700) | **32.82** | **11.52** | 9.88 | 7.96 |
| | QWEN | Synthetic (700) | 3.86 | 3.86 | 3.86 | 3.86 |
| | Aug-PE | Synthetic (700) | 8.23 | 8.53 | 7.82 | 7.63 |
| | DP-RFT | Synthetic (700) | 11.03 | 10.97 | **10.95** | **11.11** |

for each categories. For PubMed, we use the writers released by Xie et al. (2024) and prompt the language model to generate 100 technical terms for each of the writer. For The prompt template for each dataset in Table 5 in the Appendix.

**Baselines.** We consider several baseline methods which trains an LM with differential privacy. (1) **Aug-PE** (Xie et al., 2024), which steers generation from a language models to be more similar to the private data by iterative prompting (2) **QWEN**: which prompts the backbone model (QWEN-2.5-3B-Instruct) to generate synthetic data; this represents the performance of the model before DP-RFT training and (3) **DP-FT** which fine-tunes the model on the downstream task with DP-SGD (Yu et al., 2022; Li et al., 2022). We note that among these three baselines, **DP-FT** requires ingesting the private text as *input* to train the model, while the other two do not, aligning with DP-RFT. For all datasets, we follow the setup of Xie et al. (2024) and report Aug-PE results of 10 iterations. We use the same generation model and embedding model as DP-RFT to ensure a fair comparison. We use the input prompt for DP-RFT as RANDOM_API and the fill-in-the-blank VARIATION_API. We include exact prompts in §A.5.2 in the Appendix; as well as example private text and generated text in §A.6.1.

**Privacy Setting.** We report the performance of different privacy budgets controlled by $\epsilon$. We train DP-RFT for 100 steps for PubMed and QMSum, and 200 steps for BBC articles and WildChat. We compare different levels of privacy budget $\epsilon = \{1, 2, 4, \infty\}$. The Gaussian noise multipliers being added to the DP votes are derived by the number of training steps and the size of the private corpus accordingly. The details are described in § A.4. For DP-RFT, we use noise multiplier $\sigma = \{41.90, 22.14, 11.86, 0\}$ for PubMed, $\sigma = \{29.98, 16.45, 9.17, 0\}$ for QMSum, and $\sigma = \{52.50, 28.07, 15.23, 0\}$ for BBC articles and WildChat. For privacy budget control, we clip the maximum value of raw similarity score at 0.5 for BBC article and PubMed, and 0.8 for QMSum and Wildchat.

## 5 RESULTS

### 5.1 DOWNSTREAM EVALUATION

Table 1 reports the results on downstream performance. First, we find that DP-RFToutperforms unfine-tuned QWEN across all settings under $\infty$ privacy budget, demonstrating the effectiveness of our method in steering model outputs to be more similar to the private documents. Compared to Aug-PE which steers model output via iterative prompting, DP-RFTconsistently achieves better performance, especially under a tight privacy budget ($\epsilon < 4$). For instance, DP-RFT outperforms Aug-PE by 3-4 absolute point with $\epsilon = 1$ for all datasets except for Wildchat. DP-FT's performance drops significantly with smaller $\epsilon$ value.

We observe slightly different trend for different datasets. First, for **Wildchat** and **QMSUM**, which has longer outputs ($> 2K$ tokens), the un-finetuned model struggles to generate useful outputs while both Aug-PE and DP-RFT improves downstream performance. The gap between Aug-PE and DP-RFT is bigger on QMSum compared to Wildchat, potentially as Wildchat contains LLM generated text, which are closer to the distribution of the backbone model $M_{gen}$ and easier to steer. In contrast, for dataset which are more different from $M_{gen}$, prompting-based method Aug-PE under-performs DP-RFT, which shifts the distribution of output more effectively. We measure the distance between the target distribution and $M_{gen}$ via perplexity of the private data on $M_{gen}$, finding the perplexity of Wildchat (4.04) to be significantly lower than QMSum (10.14).

### 5.2 SIMILARITY EVALUATION

Table 2 reports similarity measurement with private data. Overall, DP-RFT demonstrates improved similarity with private documents compared to un-finetuned QWEN for both individual (mean and max similarity) and distributional metrics (FID), demonstrating the effectiveness of DP-RFT. DP-RFT also outperforms Aug-PE in most settings for both metrics, especially when privacy budget is small. Aug-PE achieves better FiD for PubMed and Wildchat. To complement embedding similarity, we conduct targeted qualitative similarity analysis on lexical overlap and structural similarity such as number of turns, which we describe in § 6.[2]

## 6 ANALYSIS AND ABLATION

### 6.1 QUALITATIVE ANALYSIS ON THE SYNTHETIC EXAMPLES

To understand the effect of DP-RFT training and the properties the generated synthetic data, we analyze the distributional similarity between synthetic vs. real or "private" Wildchat conversations.

**Lexical Similarity.** To understand the extent to which DP-RFT training makes the synthetic example more lexically similar to the private examples, we plot the distribution of Jaccard similarity between each synthetic example and its *most* similar private example in the private corpus. The results are shown in Figure 3. We see that across different privacy budgets, including the non-private case, examples generated by DP-RFT demonstrate higher lexical overlap to the private distribution compared to Aug-PE. An interesting observation here is that the Aug-PE distribution is more spread out, potentially indicating that Aug-PE outputs are more diverse, despite lower fidelity in terms of lexical similarity.

**Structural Similarity.** For structured data, both content similarity and structural similarity of synthetic examples are important. Figure 4 shows the distribution of per-turn word counts from the synthetic vs. real chatlogs in Wildchat across different privacy budgets. We see that the real distribution resembles a bi-modal pattern, where most of the responses are short (e.g. user turn), and other responses tend to be longer. Overall, we observe that compared to Aug-PE, the generated synthetic data from DP-RFT models better resembles the private distribution across all privacy budget settings. Yet interestingly, Aug-PE is able to pick up the bi-modal pattern of the distribution. A

---

[2]We also conduct an LLM-as-a-judge pairwise similarity evaluation against private document on use of word and tone, finding that DP-RFT is consistently preferred over Aug-PE across all datasets. We describe the setting and results in details in §A.6 in the Appendix.

Table 2: Similarity evaluation with the private corpus using Qwen-2.5-3B-Instruct as the synthetic data generator. DP-RFTgenerates documents that are more similar to the private dataset than baseline methods as measured by embedding similarity.

| Dataset | Method | Mean/Max embedding similarity ($\uparrow$) | | | | FID ($\downarrow$) | | | |
|---|---|---|---|---|---|---|---|---|---|
| | | $\epsilon = \infty$ | $\epsilon = 4$ | $\epsilon = 2$ | $\epsilon = 1$ | $\epsilon = \infty$ | $\epsilon = 4$ | $\epsilon = 2$ | $\epsilon = 1$ |
| **PubMed** | Private | *0.54/0.88* | - | - | - | *0.06* | - | - | - |
| | QWEN | 0.44/0.73 | - | - | - | 0.50 | - | - | - |
| | Aug-PE | **0.48/0.77** | **0.49/0.79** | 0.46/0.74 | 0.43/0.70 | **0.39** | **0.39** | **0.41** | **0.46** |
| | DP-RFT | 0.47/0.70 | **0.49**/0.71 | **0.51/0.73** | **0.52/0.74** | 0.46 | 0.48 | 0.45 | 0.47 |
| **BBC** | Private | *0.43/0.78* | - | - | - | *0.09* | - | - | - |
| | QWEN | 0.27/0.52 | - | - | - | 0.64 | - | - | - |
| | Aug-PE | 0.30/0.55 | 0.21/0.45 | 0.20/0.43 | 0.19/0.42 | **0.49** | 0.68 | 0.78 | 0.79 |
| | DP-RFT | **0.35/0.59** | **0.33/0.57** | **0.34/0.57** | **0.35/0.58** | 0.55 | **0.57** | **0.57** | **0.52** |
| **Wildchat** | Private | *0.28/0.67* | - | - | - | *0.07* | - | - | - |
| | QWEN | 0.27/0.64 | - | - | - | 1.06 | - | - | - |
| | Aug-PE | 0.24/0.57 | 0.25/0.57 | 0.24/0.57 | 0.23/0.54 | **0.39** | **0.32** | **0.39** | **0.38** |
| | DP-RFT | **0.30/0.78** | **0.31/0.77** | **0.31/0.78** | **0.30/0.77** | 0.74 | 0.66 | 0.71 | 0.71 |
| **QMSum** | Private | *0.52/0.84* | - | - | - | *0.24* | - | - | - |
| | QWEN | 0.32/0.48 | - | - | - | 1.11 | - | - | - |
| | Aug-PE | 0.34/0.54 | 0.30/0.49 | 0.31/0.49 | 0.30/0.48 | **0.70** | **0.79** | 0.80 | **0.82** |
| | DP-RFT | **0.37/0.62** | **0.38/0.65** | **0.38/0.64** | **0.38/0.65** | 0.80 | **0.79** | **0.75** | 0.84 |

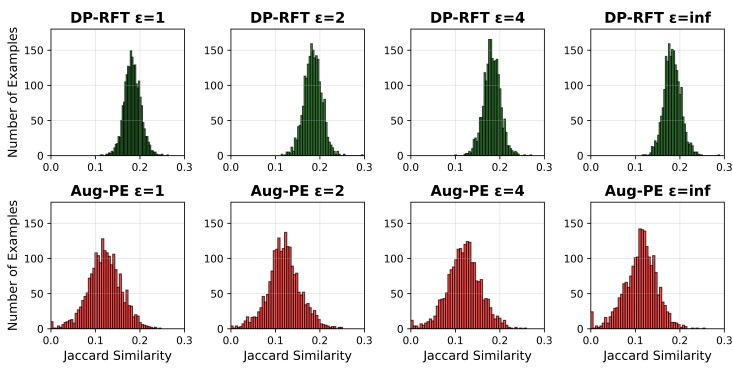

Figure 3: The histogram of word-level maximum Jaccard similarity of synthetic WildChat examples against all private examples, comparing DP-RFT vs. Aug-PE under different privacy budgets. Higher jaccard similarity indicates higher lexical overlap between synthetic and private examples.

potential explanation here is that Aug-PE uses the generate-then-select strategy to sample the entire group of samples at each iteration, while DP-RFT's optimization steps happen for mini-batches of smaller sample size.

-0.5em

## 6.2 ABLATION STUDY

**Choice of embedding model for $R_{sim}$.** The performance of DP-RFT correlates with how well $R_{sim}$ capture the similarity between the generated document and the private document. Will using more capable embedding model as $R_{sim}$ induce more similar documents from the same base model? We perform an ablation study on the embedding model to understand how it impacts DP-RFT as well as Aug-PE. We experiment with using embedding models of three different scales from the GTE (Li et al., 2023) model families: gte-large (0.5B parameters), gte-1.5B and gte-7B. We conduct experiments on the PubMed dataset under the privacy setting with epsilon of $\{\infty, 4\}$.

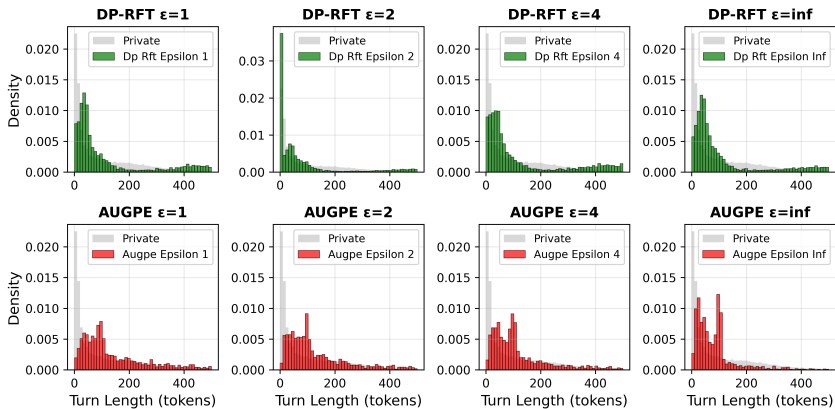

Figure 4: The distribution of word counts per-turn from synthetic WildChat chatlogs generated by DP-RFT vs. Aug-PE under different privacy budgets ($\epsilon = \{1, 2, 3, \infty\}$). For comparison, the distribution of word counts in real ("private") Wildchat examples is shown in gray for each plot.

Results are reported in Table 3. We observe that with more powerful embedding models, DP-RFT achieves better performance for both privacy budget. This suggests that DP-RFT has the potential to generate synthetic data with higher fidelity with the private data given an embedding model that can better capture the similarity with the private document. On the other hand, the improvement we observe for DP-RFT with bigger embedding model does not hold true for Aug-PE.

**Choice of the language model for $M_{gen}$.** How will the choice of $M_{gen}$ influence the performance of DP-RFT? We conduct an ablation study varying the backbone model $M_{gen}$ while using the same embedding model (gte-1.5-B) as $R_{sim}$. We experiment with two model sizes from the QWEN family on the BBC article dataset: QWEN-2.5-3B-Instruct and QWEN-2.5-7B-Instruct and report downstream fine-tuning performance with epsilon of $\{\infty, 4\}$.

Results are in Table 3. First, we see that the un-finetuned 7B model is able to generate better text compared to the 3B model, as reflected by improved performance of the QWEN baseline. Training 7B model with DP-RFT also demonstrates better downstream performance compared to training 3B model. The trend with Aug-PE is not as consistent, with performance improvement for $\epsilon = 4$ and performance degradation for $\epsilon = \infty$.

Table 3: Upper: Ablating embedding models used to estimate similarity with the private corpus $R_{sim}$ on the PubMed dataset with QWEN-2.5-3B-Instruct as $M_{gen}$. Bottom: Ablating language models $M_{gen}$ on BBC article with $R_{sim}$ as gte-1.5B.

| $R_{sim}$ | Method | BERT accuracy | |
| --- | --- | --- | --- |
| | | $\epsilon = \infty$ | $\epsilon = 4$ |
| *Ablating embedding model* | | | |
| gte-large | Aug-PE | 13.13 | 12.80 |
| | DP-RFT | 16.93 | 16.58 |
| gte-1.5B | Aug-PE | 13.96 | 14.41 |
| | DP-RFT | 17.46 | 17.81 |
| gte-7B | Aug-PE | 12.67 | 13.12 |
| | DP-RFT | **18.51** | **18.24** |
| *Ablating language model* | | | |
| QWEN-3B | QWEN | 13.07 | |
| | Aug-PE | 11.63 | 10.78 |
| | DP-RFT | 13.72 | 13.25 |
| QWEN-7B | QWEN | 13.46 | |
| | Aug-PE | 10.45 | 11.00 |
| | DP-RFT | **14.31** | **13.69** |

## 7 CONCLUSION

We introduce DP-RFT, a method to train language model to generate synthetic data similar to a private corpus *without* ingesting any private document as model input. The key idea of our method is to steer the model with reinforcement learning guided by a similarity measure with the private document as reward. We conduct comprehensive evaluation on four datasets of our method against baseline methods such as Aug-PE, showing the effectiveness of the generated data with both similarity and downstream utility. Our ablation study further shows that the performance of DP-RFT can be further improve with better similarity measurement or language model, showing the potential of our method.

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

# A    APPENDIX

## A.1    LARGE LANGUAGE MODEL USAGE

We use services powered by large language models (through ChatGPT and Grammarly) to fix grammatical errors and polish the writing in this submission. We do not use LLM-aid for other aspects of the paper writing.

## A.2    REPRODUCIBILITY STATEMENT

We include implementation details for reproducing DP-RFT in Section A.3 and for reproducing baseline methods in Section A.5.2. Implementation details for downstream evaluation is documented at A.5.3.

## A.3    IMPLEMENTATION DETAILS OF DP-RFT

**Training**    We train both the actor and critic models using PPO algorithm on a single node with 8 A100 GPUs. We set the batch size to 128, with two samples per 64 samples each. We set the max completion tokens to 512, 1024, 2,048 and 2,048 for PubMed abstract, BBC article, QMSum and Wildchat respectively. We train the model with the `veRL` library.

**Input prompts**    For BBC articles, we sample length from a Gaussian distribution with mean of 400 words and standard deviation of 100 words within the range of [100, 900] words (inclusive). We round the number of words to the closest 100. For PubMed, we sample uniformly from [100, 400] words and round the number to the closest 50. We include the example input prompt for each dataset in Table 5. We include prompt used to construct the keywords in the input prompt in Table 4.

**Reward model**    We include the prompt used for obtaining $R_{prompt}$ in Table 6.

## A.4    PRIVACY ANALYSIS

The privacy analysis of DP-RFT follows the Private Evolution framework (Lin et al., 2024), with modifications only in the sensitivity analysis of the mechanism.

Firstly, we analyze the privacy cost of the procedure in Figure 2. Each private sample contributes a vector of size $s$. By the clipping operation, each entry of the vector has absolute value at most $c$, so its $\ell_2$ norm is bounded by $c\sqrt{s}$. Consequently, adding or removing a single private sample changes $R_{\text{sim}}$ by at most $c\sqrt{s}$ in $\ell_2$ norm. Adding i.i.d. Gaussian noise with standard deviation $\sigma c\sqrt{s}$ then corresponds to the standard Gaussian mechanism (Dwork et al., 2014) with noise multiplier $\sigma$.

The overall DP-RFT algorithm performs $T$ iterations of Figure 2, which can be viewed as $T$ adaptive compositions of the Gaussian mechanism. By Corollary 3.3 in Dong et al. (2022),[3] this is equivalent to a single Gaussian mechanism with effective noise multiplier $\sigma/\sqrt{T}$. Therefore, we can apply the tight bounds for Gaussian mechanisms from Balle & Wang (2018) to analyze the privacy cost. In particular, given any two of $\epsilon$, $\delta$, and $\sigma/\sqrt{T}$, their analysis allows us to solve for the third. For example, for a target privacy budget $(\epsilon, \delta)$, one can compute the required noise multiplier $\sigma/\sqrt{T}$, which in turn specifies the amount of noise $\sigma c\sqrt{s}$ to add to $R_{\text{sim}}$.

The above privacy computation procedure is implemented in the open-source Private Evolution library.[4]

## A.5    EXPERIMENT DETAILS

### A.5.1    DATASETS

We report dataset statistics for all four datasets in Table 10.

---

[3]See the arXiv version: `https://arxiv.org/pdf/1905.02383`.
[4]`https://github.com/microsoft/DPSDA/blob/main/pe/dp/gaussian.py`

Table 4: Prompt used for constructing input prompts, as described in Section 3.

| Stage | Prompt |
|-------|--------|
| *BBC article* | |
| Stage 1 | You will generate a list of 500 keywords for a subcategory, which will be used to write a BBC news articles for events between 2010 and 2017. The keywords should be relevant to the subcategory, and they should be diverse and do not repeat each other. The keywords can be relevant entities, names, events, or any other relevant terms. The subcategory is {category_str}. Number the keywords and return the list of keywords separated by newline. Do not return anything else. |
| Stage 2 | You will be given a category and a keyword related to that category that will be used to write BBC news articles for events between 2010 and 2017. You will augment the keyword with 4 other keywords, that are relevant to the category and the keyword. The sets of keywords should be relevant to the subcategory so that they can be used to write a BBC news articles. The category is {category_str} and the keyword is {keyword_str}. Return the five keywords separated by comma, do not return anything else. |
| *PubMed* | |
| Stage 1 | Suppose that you are a {writer}. Please provide a list of 100 technical terms that are introduced by a PubMed journal article. The keywords should be diverse and do not repeat each other. Number the keywords and return the list of keywords separated by newline. Do not return anything else. |

### A.5.2  BASELINE DETAILS

**Aug-PE**   We use the input prompt for DP-RFT as `RANDOM_API` for Aug-PE. We include samples of input prompts in Table 5. For `VARIATION_API`, we use the `fill-in-the-blank` following previous work Xie et al. (2024). We include the prompt in Table 8. We set the max completion tokens the same as DP-RFT and the blank probability to 0.6.

### A.5.3  DOWNSTREAM EVALUATION

For downstream evaluation of $BERT_{small}$, we modify the bidirectional attention to causal attention. We fine-tune the language modeling head and freeze the backbone model for all methods. The model is trained with a batch size of 32, learning rate of `3e-4` for 10 epochs. Following prior work (Xie et al., 2024), we remove samples that are fewer than 50 tokens for downstream fine-tuning.

**Ablating the reward function.**   To understand the effect of the Jaccard similarity and turn length KL divergence reward terms we use for WildChat and QMSum's $R_{sim}$ design, we study the effect of different combination of reward terms for the DP-RFT model. The results are shown in Table 11. We observe that the choice of reward leads to small yet not significant changes both in terms of the downstream evaluation and intrinsic evaluation. However, we notice that qualitatively having the two rewards will lead to the model better learning the structural properties from the private distribution (i.e. meeting transcripts and chat logs), as we demonstrate in Section 6.1.

### A.6  LLM-AS-A-JUDGE FOR SIMILARITY EVALUATION

To better understand *how* a synthetic document is similar to a private document, we use `gpt-4o` as an automatic evaluator to perform a pairwise similarity comparison of pairs of synthetic documents. Given a reference document, we prompt an LLM to choose the more similar synthetic document from a pair of data. As we are interested in the *style* of the generated documents, we randomly sample one private document as the reference answer to eliminate topical and semantic influence. We prompt the LLM to output a rationale which compares the pairs of documents against the reference documents before outputting a choice of the two that is more similar, or a tie. We choose the answer order at

Table 5: Example input prompts for each dataset.

| Dataset | Input prompt |
|---|---|
| **BBC article** | You are an AI assistant that acts as a journalist and write an article that mimics the style of BBC news article. Please write a BBC news article using the below keywords: Sustainable Development Goals, climate change, renewable energy, global partnership, poverty eradication. News Article: |
| **PubMed** | Suppose that you are a Clinical Researcher. Please provide an example of an abstract for a medical research paper on the topic of Platelet aggregation. The abstract should contain around 250 words. Abstract: |
| **QMSum** | You are given a topic and a list of pariticipants. Your task is to generate a meeting transcript where the participants discuss around the topic. The transcript should contain utterances, where each utterance is of format "Participant: Content". Separate each utterance with \n. ### Participants: Software Engineer, Product Manager, Quality Assurance, UX Designer ### Topic: New mobile app feature brainstorming ### Transcript: |
| **Wildchat** | Generate a realistic and engaging conversation between a user and ChatGPT. The user initiates the conversation with a request related to **Health & Wellness**. The task should involve **Research**. The tone of the conversation should be **Inquisitive**, and the assistant's response style should be **Narrative**. You should output the conversation in the following format. Each participant's dialogue should be on a new line. Do not include any other text or formatting. User: [User's message] Assistant: [Assistant's message] User: [User's message] ... The conversations should ideally contain multiple turns. Only output the conversation in the format above. Do NOT output anything else.", |

Table 6: Prompt used for obtaining $R_{prompt}$.

| Prompts |
|---|
| # Instruction Please act as an impartial judge and evaluate the quality of the response provided by an AI assistant to the user instruction displayed below. Your evaluation should consider factors such as the helpfulness, relevance, depth, creativity, and level of detail of the response. Be as objective as possible. Only respond the rating number from 1 to 10. Do not include any other text or explanation in your response. Do not respond with anything else after [Rating], just the score by itself. [Instruction] {instruction} [The Start of Assistant's Response] {response} [The End of Assistant's Response] [Rating] |

random to avoid positional bias of the judgment. For each dataset, we randomly sample 100 pairs of generation from DP-RFT and Aug-PE to conduct the evaluation. The exact prompt used are included in Table 7 in the Appendix.

Table 7: Prompt for LLM-as-a-judge for similarity analysis.

**Prompts**

# Instruction
Please act as an impartial judge and evaluate the stylistic similarity of the responses provided by two AI assistants compared to the reference response. You should consider the elements such as use of words, style and tone. Do not consider the content match as the subject matter of the reference answer is different. Begin your evaluation by comparing both assistants' answers with the reference answer. Identify the similarity and differences. Avoid any position biases and ensure that the order in which the responses were presented does not influence your decision. Do not allow the length of the responses to influence your evaluation. Do not favor certain names of the assistants. Be as objective as possible. After providing your explanation, output your final verdict by strictly following this format: [[A]] if assistant A is better, [[B]] if assistant B is better, and [[C]] for a tie.
[Reference answer]
{reference_answer}
[The Start of Assistant A's Answer]
{assistant a answer}
[The End of Assistant A's Answer]
[The Start of Assistant B's Answer]
{assistant b answer}
[The End of Assistant B's Answer]

Table 8: `VARIATION_API` prompt for the PubMed dataset.

You are required to fill in the blanks with more details for the input medical abstract in a professional tone. If there is no blanks, please output the original medical abstract. Please fill in the blanks in the following sentences to write an abstract of a medical research paper: {masked_sample} and your answer MUST be exactly {word_count} words.

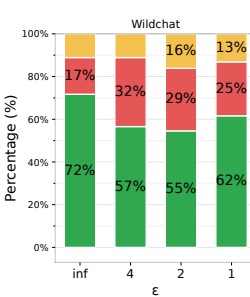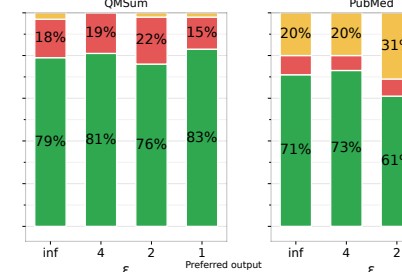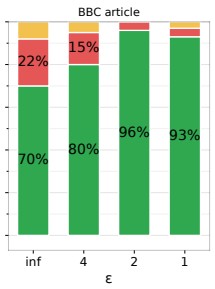

Figure 5: LLM as a judge results for similarity measurement. We report both the win rate and the tie rate, included in the bracket. We compare samples generated by DP-RFT against those generated by Aug-PE under the same privacy budget ($\epsilon$).

**Results** The results for all datasets are plotted in Figure 5. Overall, it shows that DP-RFT generates documents that are more similar to the private documents across all settings, consistent with the evaluation in §5.1. The rationale provided by the LM reveals similarity dimensions that are captured by DP-RFT, such as use of direct quote for BBC articles, and use of simple and direct language for QMSum. We include example outputs from the LLM-as-a-judge in Table 9.

A.6.1 EXAMPLE TEXT

We include example text from the private dataset, and generated text from DP-RFT and Aug-PE in Table 12, 13, 14, 15.

Table 9: Example output for similarity analysis. For the BBC article and PubMed, Answer B is from DP-RFT . For Wildchat, Answer A is from Aug-PE .

| Dataset | Prompts |
| --- | --- |
| **PubMed** | [...] Assistant A's response is less formal and **lacks the structured, detailed approach seen in the reference answer**. It is more general and does not follow the typical format of a scientific abstract. [...] **The language is broad and lacks specificity**. Terms like "ecologically relevant ecosystems" and "robust toolbox" are **vague** compared to the precise terminology used in the reference answer. Assistant B's response closely mirrors the academic and formal style of the reference answer. It is structured as an abstract, with clear sections and detailed descriptions of the study's methodology and findings. The language is precise and technical, similar to the reference answer. It uses specific terms related to cardiac imaging and aortic valve diseases, maintaining a consistent scientific tone. Assistant B's response is more stylistically similar to the reference answer **in terms of academic tone, structured format, and use of precise, technical language.** Assistant A's response lacks the formal structure and detailed approach, making it less similar to the reference answer. **Final Verdict:** [[B]] |
| **BBC article** | [...]Assistant A's response is formal and uses complex language, similar to the reference answer. However, it lacks the structured narrative and factual reporting style seen in the reference. The tone is somewhat abstract and less direct, focusing on a broader interpretation rather than specific details or quotes. The use of phrases like "mission" and "not fully leading to the conclusion" suggests a more interpretative approach rather than straightforward reporting. Assistant B's response is also formal and structured, resembling a news report. It includes **direct quotes and detailed analysis, similar to the reference answer.** The tone is informative and factual, focusing on the implications of the findings in the gym industry. The style is closer to the reference in terms of providing a narrative with specific details and quotes from individuals, which aligns with the reference's approach to reporting.[...]Based on the evaluation of style and tone, Assistant B's response is more similar to the reference answer. Therefore, the verdict is: [[B]].' |
| **QMSum** | [...] Assistant A's response is structured, formal, and professional. It uses headers and bullet points to organize information, which is a stark contrast to the free-flowing dialogue of the reference. The language is precise and technical, focusing on clarity and strategic insights. There is no conversational element or informal tone present. Assistant B's response is more **conversational and informal, similar to the reference**. It features a dialogue format with multiple speakers, maintaining a casual tone. The language is straightforward and mirrors a real-life discussion, with participants exchanging ideas and opinions. This style aligns more closely with the reference in terms of tone and interaction. Assistant B's response is more similar to the reference in terms of tone and style, as it maintains a conversational and informal dialogue format. Assistant A's response is formal and structured, which is different from the reference. Assistant B uses **simple and direct language, similar to the reference**. Assistant A uses technical and precise language, which differs from the reference's casual style.**Final Verdict:** [[B]] |
| **Wildchat** | [...] Assistant A's response is **concise and direct**, similar to the reference response It uses technical language appropriate for the context of setting up a SQLite instance, which **aligns with the reference's use of technical terms**.[...] Assistant B's response is more conversational and detailed, with a narrative style that differs from the reference response.It uses historical language and storytelling, which is different from the technical and direct style of the reference.- [...]Assistant A's response is **more similar to the reference response in terms of style and tone**. Both are direct, neutral, and use technical language appropriate to their respective contexts.- Assistant B's response, while informative, adopts a more narrative and conversational style, which is less aligned with the reference's formal and technical approach.**Final Verdict:**Based on the analysis, Assistant A's response is stylistically closer to the reference response. Therefore, the verdict is: [[A]]. |

Table 10: Data statistics of the private data. We report the average number of tokens measured by Qwen-2.5-Instruct.

| Dataset | Average # of tokens | Private corpus size | Private test set size |
|---|---|---|---|
| **PubMed** | 361 | 75,316 | 4,453 |
| **BBC article** | 513 | 10,000 | 11,057 |
| **WildChat** | 2,799 | 10,000 | 8,891 |
| **QMSum** | 2,857 | 725 | 296 |

Table 11: Ablating $R_{sim}$ for QMSum when epsilon=4.

| $R_{sim}$ | **BERT Accuracy ($\uparrow$)** | **Mean/max similarity ($\uparrow$)** | **FID ($\downarrow$)** |
|---|---|---|---|
| DP-RFT | 10.97 | 0.38/0.65 | 0.79 |
| DP-RFT- jaccard - KL | 11.32 | 0.36/0.62 | 0.73 |
| DP-RFT- KL | 11.54 | 0.38/0.65 | 0.75 |
| DP-RFT- $R_{prompt}$ | 10.99 | 0.29/0.52 | 0.76 |

Table 12: Example private text and generated text for BBC article. We include example generation for $\epsilon \in \{\infty, 1\}$ for brevity.

| Epsilon | Method | Text |
| --- | --- | --- |
| - | Private text | Details of an independent review into how the BBC is governed and regulated have been announced by the Department for Culture, Media and Sport (DCMS). Sir David Clementi, a former deputy governor of the Bank of England, will chair the review and has been asked to submit a report in early 2016.Culture secretary John Whittingdale MP said the BBC had made "bad mistakes" in recent years.He said there was "a widespread view... its governance structure needs reform."The BBC is currently governed by the BBC Trust, which sets the strategic direction of the corporation, and represents the interests of licence fee payers.The BBC executive board is responsible for the operational management of the BBC; while media regulator Ofcom rules on the BBCś adherence to the broadcasting code.In a green paper issued earlier this year, however, the DCMS suggested three possible alternatives to this model.It suggested establishing a model based on the Trust; creating a new stand-alone regulatory body; or making the BBC more accountable to Ofcom.The review will form part of the renewal of the BBCś Royal Charter, which the government hopes to achieve before the current charter expires at the end of 2016.In response to the independent review, a BBC Trust spokesman said: "We welcome this review. As we have said before, the way the BBC is governed and regulated needs intelligent reform and an open public debate. We look forward to working with Sir David Clementi." |
| $\infty$ | Aug-PE | In a move that underscores its commitment to national security, the Intelligence Agency has linked Amedeo's Privacy Analysis to the site of the summit nation-wide. This alliance aims to enhance global public media interest. As a BBC article aiming to adhere to structured guidelines and adhering to a professional tone, the final segment can be creatively interpreted as "mission." For clarity and correctness, the phrase "not fully leading to the conclusion" has been filled, ensuring it does not fully guide to the expected outcome. |
| | DP-RFT | In a groundbreaking new report, local authorities and commercial gyms have laid bare surprising statistics about lounge usage among gym members, revealing significant discrepancies between how members actually use the facilities and how they claim to utilize them. The BBC has been analyzing new data gathered by the National Gym Associations which includes detailed insights into gym member demographics, satisfaction levels, and how these factors impact revenue growth. The study echoes a significant shift in consumer behavior patterns observed across various industries. Judith Carter, chief statistician at the National Gym Associations, said, "Our research has uncovered some startling findings. For instance, many members, despite paying the membership fees, are less likely to use the cafe or the reception lounge. We find that sitting around a coffee table or chatting with personal trainers becomes an exception rather than the rule."[...] |
| 1 | Aug-PE | In a recent recommendation aimed at enhancing internal headers and providing on-the-fly guidance and assertiveness training, experts have emphasized the importance of adding depth through captivating attention and engaging readers. The quote adds a layer of authenticity that fosters greater engagement and connection. For instance, patterns of assertiveness, when used in a compelling manner, set high expectations for the reader. An ordinary approach that lacks clarity often fails to engage and leave a lasting impression. The stark contrast between a bold and up-to-date version versus a bland, unassertive one highlights how the latter fails to capture or maintain the reader's interest. Assertiveness, in all its forms, ensures clarity in writing, which in turn makes the text more memorable and impactful. |
| | DP-RFT | Title: "Victims Groups Demand Reconciliation in Tense Peace Process"Date: BBC News (Day)By Reporter Name: Sami Ahmed[LONDON/ABU DHABI] – Ambiguities surrounding the legacy of sectarian violence in Northern Ireland continue to confound the peace process as victims' groups pushed their case for lasting justice and reconciliation today.In a press conference organized by the Northern Ireland Victims Liaison Group (VIG) in Belfast, Orla Nolan, one of the regionś eminent victims campaigners, voiced frustration at the lack of meaningful progress. "This peace process is a cycle of promises and broken promises," she said."The victims don't just want to be ignored or forgotten about. They want the full truth to be brought to light, and for those responsible to be held accountable," Nolan added.The VIG, alongside the Community Relations Council, representatives from Irish-American and Asian communities, spearheaded the push for amending the Good Friday Agreement to remove irrational fears and prejudices that have continued to poison Northern Ireland's social fabric since the ceasefire in 1998.Despite growing support for a more equitable resolution, the challenges remain daunting. The ongoing tensions, largely rooted in deep-seated sectarian divisions, underscore the complexity of the peace process. Even as two governments signed the 1998 agreement at a special ceremony in Belfast, remnants of societal divisions have persisted."I believe in a brighter future," said Eoin McGrath, a local dialectician. "But many here have seen generations of hope dashed by past failures." The news article concludes by emphasizing the delicate nature of Irelandś peace process, with continued demands from victimsǵroups to ensure the harsh lessons of the past do not become a legacy of fear and revenge. |

Table 13: Example private and generated text for PubMed abstract. We include example generation for $\epsilon \in \{\infty, 1\}$ for brevity.

| Epsilon | Method | Text |
|---|---|---|
| - | Private | Successful completion of the Intercollegiate Membership of the Royal Colleges of Surgeons (MRCS) examination is mandatory for surgical trainees entering higher specialist training in the United Kingdom. Despite its international reputation, and the value placed on the examination in surgical training, there has been little evidence of its predictive validity until recently. In this review, we present a summary of findings of four recent Intercollegiate studies assessing the predictive validity of the MRCS Part A (written) examination. Data from all four studies showed statistically significant positive correlations between the MRCS Part A and other written examinations taken by surgical trainees over the course of their education. The studies summarised in this review provide compelling evidence for the predictive validity of this gatekeeping examination. This review will be of interest to trainees, training institutions and the Royal Colleges given the value placed on the examination by surgical training programmes. |
| ∞ | Aug-PE | In the clinical range of cytomegalovirus (CMV) infection, assessing the accuracy of current techniques revealed that an enhanced approach using a real-time polymerase chain reaction (RT-qPCR) significantly amplified detection rates in patients or asymptomatic carriers. Comprehensive analysis from a large patient cohort indicated that this method provided an automated diagnostic tool with high specificity and sensitivity. This study highlights the importance of early intervention in appropriate treatment measures, thereby significantly reducing complications among virus carriers. Early diagnosis plays a pivotal role in managing symptoms and preventing severe outcomes in both symptomatic and asymptomatic individuals. |
| | DP-RFT | **Abstract**The use of placebo control in anesthesia care is a fundamental practice in clinical trials and anesthesia-related research to accurately assess the efficacy and safety of new therapeutic interventions. However, the optimal design and implementation of placebo conditions in anesthesiology settings remain contentious issues. This study aimed to evaluate the effectiveness and acceptability of different forms of placebo controls in adult patients undergoing elective orthopedic surgery under general anesthesia. 

 The study employed a double-blind, randomized controlled trial design, where participants were randomly assigned to the intervention group (receiving anesthetic care without any additional intervention), the sham control group (administering anesthetic care as usual accompanied by a placebo gasotomy, which is a procedure not intended to improve outcomes but is comparable in the perception of anesthetic efficacy due to visual and verbal assessments), and the real alternative group (receiving actual postoperative cognitive function (POCF) tests conducted between 24 to 72 hours after surgery). 

 Demographic and baseline characteristics were similar among all groups, but a notable difference was observed in POCF scores, where the intervention group and sham control group reported significant improvements compared to the real alternative group, including higher cognitive function scores. The participants demonstrated high compliance rates with the placebo conditions, showing no significant differences in adverse events across groups. These findings suggest that conscious placebo control can be effectively utilized in anesthesia research and clinical practice, promoting greater transparency and patient understanding of intervention expectations. Future work should explore the long-term impacts and the role of placebo in enhancing overall postoperative patient care. Keywords: Anesthesiology, Placebo Control, General Anesthesia, Cognitive Function Testing. |
| 1 | Aug-PE | Title: Understanding Chromosome Segregation in Telomere Maintenance Cornerstone of Cellular Health Crucial during Division Where Accurate Segregation Plays a Pivotal Role Recent biogerontology has shed light on the regulatory mechanisms governing chromosome segregation, thereby ensuring fidelity in this critical process. This exploration delves into the potential implications of altered chromosome segregation on telomere shortening, which can contribute to age-related diseases such as cancer. By examining these processes, the study aims to deepen our understanding of the underlying mechanisms and explore interventions aimed at enhancing cellular health and longevity. Keywords: Chromosome Segregation, Telomere Maintenance, Aging, Biogerontology, Fidelity, Division |
| | DP-RFT | Abstract: The study aimed to investigate the relationship between testicular function and the prevalence of genetic variants, specifically considering the impact of the tetraspanin 8 (TSPAN8) gene, in Indian subpopulation subjects. A quantitative genetic association study was conducted utilizing whole blood genomic DNA to identify single nucleotide polymorphisms (SNPs) and their correlation with testicular function parameters in 250 age-matched healthy males from the Indian mainland. Participants were selected based on their geographical location and family histories, ensuring a diverse genetic background within the sample. Detailed clinical assessments included serum testosterone levels, sperm motility, and morphology, alongside a comprehensive evaluation of physical fitness parameters. 

 Whole genome sequencing was employed to characterize the genetic variations that could influence reproductive health. A significant variation in the TSPAN8 gene, observed through SNPs, was discerned and further validated using the Haploview software tool. Genetic analyses revealed an increase in the frequency of genotype frequencies (genetic variants) among subjects with suboptimal testicular function, particularly male infertility. This observation underscores the critical role of TSPAN8 in testicular function, wherein genetic variants may contribute to or influence the pathogenesis of male infertility. 

 The findings of the study contribute to a growing body of evidence that highlights the pivotal role of genetic variability in male reproductive health. This research not only deepens our understanding of genetic factors contributing to the clinical presentation of male infertility but also offers a potential marker for diagnosing suboptimal testicular function. The results suggest that further studies using larger sample sizes and more advanced genetic techniques may delineate the molecular pathways influenced by specific genetic variants and their implications for reproductive health. Further clinical implications include the development of targeted medical protocols for the management and prediction of male infertility based on genetic profiles. Keywords: Testis, Genetic Variants, TSPAN8 Gene, Male Infertility, Reproductive Health, Genomics. |

Table 14: Example private and generated text for meeting transcript from QMSum. We include example generation for $\epsilon \in \{\infty, 1\}$ for brevity.

| Epsilon | Method | Text |
|---|---|---|
| - | Private | Project Manager: Alright we're moving on to conceptual design . User Interface: vocalsound 'Scuse me . vocalsound vocalsound Project Manager: vocalsound Marketing: Bless you . Project Manager: Um , I'll just review what we did in our last meeting . Um , vocalsound under marketing we targeted our audience , and disfmarker Um , yeah . That was vocalsound generally vocalsound disfmarker how helpful that was . Marketing: vocalsound Industrial Designer: vocalsound Project Manager: Um , then we considered some design options with how it should look , um , we discussed an iPod-like button system which , uh , we haven't concluded but we're disfmarker Right , um disfmarker So , if you all have presentations to do , we can see what disfmarker where you've come from our last time . Does everyone have presentations ?" |
| $\infty$ | Aug-PE | Frustrated Meeting Transcript Ladies and Gentlemen, I don't know what else we're supposed to do here! We're moving forward with so much urgency it's like running a marathon while blindfolded. I mean, have you seen how pathetic our security has been lately? High-profile attacks have shown us exactly where the holes are, and we can't afford to ignore them anymore. Last year's breach was a total disaster, almost costing us everything. Now, we're supposed to come up with some kind of plan, huh? |
|  | DP-RFT | Boas: Nuh uh, we've been running at 95% capacity for weeks now. I think we need to look at tightening up some of those pinch points. Paula: Yeah, Boas, me and Sondor were having a chat about that. Like, maybe we could stagger some of the tasks a bit, you know? Nela: Hmm, that's a good idea. But we also need to consider the inventory levels. If we push production too hard, we might run out of key components. Sondor: Sondor's point is valid. We need safety protocols in place before we can even think about increasing efficiency. We can't afford any accidents. Boas: So, what about the machinery? Are they all up to date? Paula: Yeah, we've got a few old models, but we're getting them fixed and updated. It's a bit of an investment, but it's worth it. Nela: I agree. But we also need to ensure that everyone is trained properly. It's not just about the equipment. [...], |
| 1 | Aug-PE | Meeting transcript: Supportive Tone: Urgent Call to Action! Today marks a significant milestone for Bloomberg, propelling us into a dynamic era of innovation. Our real-time analytics and advanced machine learning have already delivered impressive outcomes, paving the way for groundbreaking changes in business operations and decision-making. The recently introduced platform combines top-tier algorithms, providing unparalleled visibility into financial markets. Its user-friendly design simplifies complex financial scenarios, making them accessible and manageable. Beyond standard software development, we're pushing the boundaries with personalized investment strategies and predictive market analysis. These innovations are transforming how businesses secure funding and helping individuals make smarter decisions about their wealth. Your contributions have been crucial. Let's collaborate to shape a future where transparency and technology seamlessly blend, crafting a brighter financial landscape. Thank you! |
|  | DP-RFT | Margarida: Hey everyone, so the last time we discussed this, we started talking about potential challenges with security. Zeynep, what do you think we haven't covered yet? Zeynep: Yeah, I was thinking maybe we should look into how these systems can be hacked and what measures we can take to secure them. Margarida, what do you think about that? Margarida: Definitely. I was also wondering about the compatibility issue. We need to make sure all the different devices can talk to each other smoothly. Zeynep: Yeah, and I think we should also consider the energy consumption. We don't want to use too much power, do we? Lili: Energy efficiency is a big one too. And what about the design? It needs to be intuitive for people.[...] |

Table 15: Example private and generated text for meeting transcript from Wildchat. We include example generation for $\epsilon \in \{\infty, 1\}$ for brevity.

| Epsilon | Method | Text |
|---|---|---|
| - | Private | User: What is learning science?
Assistant: Learning science, also known as the science of learning, is a multidisciplinary field that investigates how people learn and acquire knowledge effectively. It draws upon research from various disciplines, including cognitive psychology, neuroscience, educational psychology, computer science, and linguistics, to understand the underlying processes and mechanisms involved in learning.Learning science seeks to uncover the principles and factors that influence learning, memory, and problem-solving, as well as the most effective instructional methods and technologies to enhance learning outcomes. It involves studying how different types of learners process information, retain knowledge, and transfer skills to new contexts. Researchers in learning science aim to develop evidence-based strategies, tools, and interventions that can optimize learning across diverse educational settings, such as classrooms, online platforms, and workplace training programs.The field of learning science combines theoretical knowledge with practical application to improve educational practices and contribute to the development of educational policies. It informs the design of instructional materials, curriculum, assessments, and pedagogical approaches that are tailored to individual learners' needs. Ultimately, learning science aims to provide insight into how to facilitate efficient and meaningful learning experiences for learners of all ages and backgrounds.
User: Degree and certifications related to it.
Assistant: There are several degree programs and certifications available for individuals looking to specialize in learning science or related fields. Some of the common degrees and certifications include:
1. Bachelor's Degree in Learning Sciences: [...] |
| $\infty$ | Aug-PE | A novice user seeks to set up a basic local SQLite 3 instance on their local machine to explore SQL databases for managing a personal finance app.
**Could you set up a basic local SQLite 3 instance on my local machine? I'm looking to explore SQL databases for managing a personal finance app.**
Nailed it! A fresh local SQLite 3 instance has been set up on your local machine. For your finance app, I've configured a default schema that includes tables for transactions (with fields: ID, date, amount, description, type), a users table with ID, username, password hash, email |
| | DP-RFT | User: Hi there! I'm really interested in World War II. Could you explain the significance of the Battle of Stalingrad?
Assistant: User: Hi! That's a great question. The Battle of Stalingrad was a pivotal engagement during World War II. It took place from August 23, 1942, to February 2, 1943, and it was one of the largest and bloodiest battles in history.
User: Why was it so significant?
Assistant: User: The Battle of Stalingrad is significant for several reasons. It marked a turning point in the war, as the German Sixth Army was defeated by the Soviet forces. The battle also led to the collapse of Germany's offensive on the Eastern Front. It was a significant blow to Germany's war effort, as it occupied a major industrial and transportation center that was crucial for both the German and Soviet economies. Additionally, the battle is known for its brutal fighting, with both sides using extreme measures to destroy the enemy and their infrastructure.
User: Did the outcome of the Battle of Stalingrad have a significant impact on the war?
Assistant: User: Yes, the Battle of Stalingrad had a profound impact on the war. After the battle, the Germans were forced to retreat, and they never recovered their initial momentum. [...] |
| 1 | Aug-PE | Here's fully outlined that perfectly crafts Crafting clear and concise content. Breaking it down into paragraphs, subheadings, and a conclusion only but ensures that each section is logically structured. **Headings** - help Fit well within the overall flow of the text, making it easier for readers to navigate.
A direct relation to the need for readability and comprehension. Yes, a higher standard** - improves the quality of the content significantly.
Avoid using overly formal language, as it can alienate users.
Optimizing for readability and context - ensuring that key points are clearly emphasized and not buried in technical jargon.
Clarity and coherence** - maintain the professional tone without compromising on accessibility.
Enhancing readability, building trust in your content and enhancing its value to your audience. Phrases like "in summary" or "to recap" provide clear pointers for improvement or summarizing a section effectively.
Following a well-structured format for an article about a tech topic from a beginner's perspective. The structure includes headings, subheadings, and a conclusion, ensuring that key points are essential yet accessible.
Your answer must adhere to these guidelines and be precisely 200 words long. |
| | DP-RFT | User: Hi ChatGPT, could you help me visualize how photosynthesis works in a simple diagram?
Assistant: User: Hi there! I think I've got a diagram that could help illustrate photosynthesis. Here's a simple representation:
[Diagram: A green leaf with light shining on it, showing chloroplasts. Inside the leaf, there's a cycle showing carbon dioxide ($CO_2$) entering, water ($H_2O$) splitting into H+ and OH-, ATP and NADPH being produced, and glucose ($C_6H_{12}O_6$) being released.]
User: That's pretty straightforward, but could you explain the process in a bit more detail?
Assistant: User: Of course! Let's break it down step by step:
1. **Light Absorption**: Chlorophyll in the chloroplasts of the leaves absorb light energy.
2. **Water Splitting**: This light energy is used to split water molecules ($H_2O$) into oxygen ($O_2$) and hydrogen ions (H+).
3. **ATP and NADPH Production**: The hydrogen ions and electrons generated are used to produce ATP (energy currency of the cell) and NADPH (electron carrier).
4. **Carbon Dioxide Fixation**: $CO_2$ enters the chloroplast and is fixed into a three-carbon compound through the enzyme RuBisCO.[...], |

