# OpenReview forum: "DP-RFT: Learning to Generate Synthetic Text via Differentially Private Reinforcement Fine-tuning"
_ICLR.cc/2026/Conference — Submitted to ICLR 2026_

### Official Review · Reviewer_XTBv · 2025-10-30

**Soundness:** 2
**Presentation:** 2
**Contribution:** 2
**Rating:** 2
**Confidence:** 4

**Summary:**

This paper proposes Differentially Private Reinforcement Fine-Tuning (DP-RFT), a new method for generating synthetic text from a private corpus without directly exposing the private data ("eyes-off") to the language model. The core idea is to use reinforcement learning (specifically, PPO) to fine-tune a large language model (LLM). The reward signal is derived from differentially private (DP) nearest-neighbor votes from the private corpus. This reward is based on the embedding similarity between synthetic samples and the private data, modified from prior work (Aug-PE) to use raw similarity scores instead of binary votes . To prevent reward hacking, this similarity reward is combined with a prompt-adherence reward generated by an LLM-as-a-judge. The authors evaluate DP-RFT on four datasets (news, transcripts, medical abstracts), claiming it significantly outperforms baselines like DP-SGD and Aug-PE in terms of synthetic data fidelity and downstream utility

**Strengths:**

- **Clear Problem Formulation:** The paper addresses a significant and practical problem: generating high-fidelity synthetic text from a private corpus under "eyes-off" constraints, where models like DP-SGD (which require direct data access) are not viable.
- **Solves a Key Baseline Limitation:** The paper correctly identifies a key weakness in the "eyes-off" baseline Aug-PE: because the LLM is frozen, it has "limited steer-ability" and is bounded by the quality of the off-the-shelf model's outputs. DP-RFT directly addresses this by fine-tuning the model, allowing it to adapt its weights to the private distribution.

**Weaknesses:**

- **Ad-hoc Reward Engineering:** The method relies on dataset-specific reward engineering, such as adding special KL divergence terms for the WildChat and QMSum datasets, suggesting the base embedding-similarity reward is insufficient on its own.
- **Weak Downstream Evaluation:** The claim of "downstream utility" is unsubstantiated. The evaluation only uses next-token prediction, which is a weak proxy, instead of testing on actual, relevant tasks like classification (for BBC News) or summarization (for QMSum).
- **Incremental Contribution:** The paper's novelty is limited. It presents a straightforward combination of standard PPO with a DP embedding-similarity reward, which is a minor modification of prior work (Aug-PE) and conceptually similar to other recent RL-based method Hou et al. (2025), which used DPO (an offline RL algorithm) with embedding similarity as a reward.
- **Overstated "Eyes-off" Premise:** The "eyes-off" claim is misleading. While the LLM trainer avoids direct data access, the reward calculator requires continuous, computationally expensive, and security-sensitive access to the entire private corpus's embeddings at every single training step, a significant practical limitation that is not discussed.

**Questions:**

1. **Missing Related Work and Baselines.** A potentially significant piece of related work [1] appears to be missing. This work also deals with synthetic text generation for training LLMs, though via gradient matching. Given that both methods aim to create synthetic data for LLM training, would this work not constitute a relevant baseline or, at minimum, require discussion in your related work section?

    [1] Nguyen, Dang, et al. "Synthetic text generation for training large language models via gradient matching." (arXiv:2502.17607, 2025)?

2. **Reward Hacking:** The $R_{prompt}$ term is introduced to mitigate "reward hacking"37. What happens if you remove it entirely and only use $R_{sim}$? How severe is the reward hacking? Does the model simply output copies or near-copies of the private data embeddings?
3. **On Ad-hoc Rewards:** For QMSum and WildChat, you added KL-divergence-based reward terms.

    a. Did the base method (using only embedding similarity) fail for these datasets?

    b. Could you provide an ablation (like in Table 11, but for the main results) showing the impact of *not* including these terms?

    c. How is a practitioner supposed to know which ad-hoc terms to add for a new dataset?

4. **On Downstream Utility:** Can you provide evaluation results for more meaningful downstream tasks beyond next-token prediction? For example, please show results for classification on BBC News or summarization on QMSum, as this would be a far more compelling case for "downstream utility."
5. **Member inference attack**. Have you considered evaluating the robustness of your synthetically generated data against member inference attacks?
6. **Cross-model transferability**. Is the synthetic dataset transferrable across different language model?
7. **Computational Cost:** What is the **computation time** and general computational overhead of the DP-RFT training process compared to the baselines like Aug-PE (which is iterative but does not fine-tune) and standard DP-FT?

---

> ### Author Response · Authors · 2025-11-26
>
> Thank you for your careful review! Please see our response below:
> [W1] Reward engineering:
> * The reviewer notes that our method requires dataset-specific reward engineering (e.g., adding KL-divergence terms for structured datasets like WildChat and QMSum). While true, we view this as a strength rather than a limitation. Our approach depends on having a suitable similarity reward $R_{sim}$ that can effectively capture similarity with private examples, and we demonstrate multiple instantiations of $R_{sim}$ across different types of data. As with RLVR, different domains naturally call for different reward formulations. Identifying the best $R_{sim}$ is not the focus of our work; our contribution is a general framework that trains models effectively with a reliable $R_{sim}$. We will add this as a discussion of our paper.
>
> [W2, Q6] Downstream evaluation:
> * Please refer to our general response on our downstream evaluation setting, and new results on evaluating with GPT-2 ("Clarification of our downstream evaluation set-up").
>
> [W3] Contribution and ablation:
> Please refer to our general response for clarification on our contribution ("Clarification of our contribution"). Regarding comparison with the recent work Hou et al., 2025 (POPri), we highlight several differences between their work and our work:
> * POPri proposes to use DP voting to create positive and negative pairs and then train the model on DPO. Instead, we propose to leverage the DP voting as a direct reward for online reinforcement learning methods such as PPO.
> * While POPri mostly focuses on domain specific texts (e.g. PubMed, Congressional transcripts), we also experiment with generating synthetic text for private, structural data such as meeting transcripts and chat logs. We propose similarity metrics beyond embedding similarity for these datasets.
> * Moreover, POPri focuses on federated learning while we focus on the centralized setting.
>
> [W4]  Privacy setting:
> * Please refer to our general response for clarification on our privacy setting ("Clarification on our privacy setting").
>
> [Q1] Thank you for bringing up this recent paper! As the reviewer mentioned, this paper uses gradient matching to generate privacy preserving synthetic data. Accessing model gradients from private data is a violation of the “eyes-off” setting we study in the paper. We have added a discussion of this paper in our related work section.
>
> [Q2] $R_prompt$ is proposed to prevent reward hacking on similarity w/ the embedding model with degraded quality, especially as embedding models are known to have certain bias, as we discuss in Section 3.2 of the paper. We provide an ablation study where we remove R_prompt, please refer to our general response (“Ablation on removing
> ”) for details.
>
> [Q3] Table 11 in our Section A.5.3 provides an ablation of not including the KL terms for QMSum. While including the structural similarity (DP-RFT v.s. Embedding only) leads to small yet insignificant changes both in terms of downstream evaluation and similarity evaluation, we do observe qualitatively in Section 6 that DP-RFT with the structural similarity as reward is able to learn the structures of the private documents.

---

> ### Author Response · Authors · 2025-11-26
>
> [Q5] The reviewer was curious about the membership inference attack (MIA) analysis of our method. Here, we conduct an MIA evaluation of the downstream model fine-tuned with various methods. We use the perplexity threshold of the downstream fine-tuned model to predict membership following prior work [0] and report AUC. Results show that our method achieve similar performance for robustness of MIA attack compared to Aug-PE.
>
> [0] [Carlini et al. Extracting training data from large language models. USENIX Security 2021](https://arxiv.org/abs/2012.07805).
>
> | **PubMed** | **AUC – PPL (epsilon=4)** | **AUC – PPL (epsilon=2)** | **AUC – PPL (epsilon=1)** |
> |------------|---------------------------|----------------------------|----------------------------|
> | Aug-PE     | 0.533                     | 0.528                      | **0.528**                      |
> | DP-RFT     | **0.524**                     | **0.524**                      | 0.529                      |
>
> [Q7] The reviewer raises a good point on comparing computational cost for the different methods (DP-SGD, Aug-PE and DP-RFT), which is an important axis to consider for downstream application. First, we note that both Aug-PE and DP-RFT require more time than DP-SGD, introduced by the computational time required for generating the synthetic data. While Aug-PE does not involve model training, the iteration also requires substantial computational cost (generating the variation of the text), especially for generating longer synthetic text. Below, we benchmark the time required to generate synthetic data by DP-RFT and Aug-PE for Wildchat. Note that DP-RFT’s training is a one-time cost. We report the
> wall-clock GPU hours needed for each method in our experiment settings with QWEN-3B as the data generator:
> * DP-RFT training (200 steps): ～5 hours with 8*A100 GPUS = 40 A100 hours
> * DP-RFT inference (2000 samples): ~1 A100 hour w/ VLLM inference (+prefix caching)
> * Aug-PE (2000 samples, 10 iterations): Each iteration takes ~10 hours to generate using the official AugPE implementations on 1 A100 GPUS. Running 10 iterations = 100 A100 GPU hours.
>
> Overall, we note that DP-RFT does not incur more computational time compared to Aug-PE.

---

### Official Review · Reviewer_3Y1p · 2025-11-02

**Soundness:** 2
**Presentation:** 3
**Contribution:** 2
**Rating:** 2
**Confidence:** 4

**Summary:**

The paper introduces Differentially Private Reinforcement Fine-Tuning (DP-RFT), which uses LLMs to generate synthetic data using online reinforcement learning. DP-RFT uses nearest-neighbor votes from a private corpus as a reward signal for RL. The LLM iteratively learns to generate synthetic data to maximize the expected DP votes through Proximal Policy Optimization (PPO).

**Strengths:**

- The problem is well-motivated and important for the community.
- The paper is well-written and is easy to follow.

**Weaknesses:**

- The most important way to evaluate the performance of the proposed method is to show that it provides a better "performance" compared to (1) existing eyes-off methods and (2) methods that require direct access to raw private data during training. However, in line 244 it is mentioned that the performance evaluation is done for fine-tuning BERT_small, while the synthetic data is generated with Qwen with the help of GPT-4o (line 256)? If so, I'm very surprised of this evaluation and don't see the justification. If we have access to Qwen and GPT for data generation, I'd expect that the performance evaluation is also done by fine-tuning a much stronger model. Even if this model is used in some prior work, I don't think performance of BERT_small is a good indicative of the quality of the synthetic data for any modern application.

- The similarity of the synthetic data to private data is not as important if we don't have a good performance from fine-tuning the model on synthetic data. While this is good to show, it's not enough and shouldn't be used as the primary evaluation metric. For LLMs, one can generate synthetic data close to real data with very poor performance.

- The main novelty of the paper is using similarity scores between synthetic data (generated using existing methods) to steer a model with RL. Other than this, I didn't find major novelty in the paper and to my understanding, the effectiveness of the proposed approach is not really verified (as I explained above).

**Questions:**

- Can you evaluate the performance of the proposed method when fine-tuning modern LLMs, such as Qwen, Llama, etc?

ps: I'm happy to revise my score if I misunderstood anything.

---

> ### Author Response · Authors · 2025-11-26
>
> Thank you for your review!
>
> Please refer to our general response on downstream evaluation [W1, W2, Q1] ("Clarification of our downstream evaluation set-up:"); and clarification of the novelty of our work ("Clarification of our contribution") [W3].

---

### Official Review · Reviewer_6sPe · 2025-11-03

**Soundness:** 2
**Presentation:** 3
**Contribution:** 2
**Rating:** 4
**Confidence:** 3

**Summary:**

The paper presents DP-RFT to train an LLM using reinforcement learning (RL) guided by a differentially private reward signal. This reward signal is derived from "nearest-neighbor votes" from a private corpus, but crucially, the LLM itself never directly accesses or "ingests" the raw private data during training (an "eyes-off" approach). This addresses a critical trade-off: traditional private fine-tuning (like DP-SGD) requires direct access to private data, while existing "eyes-off" methods often produce synthetic text lacking domain fidelity.

**Strengths:**

S1. DP-RFT integrates differentially private reinforcement fine-tuning (RFT) to train large language models (LLMs) for synthetic data generation. I like its "eyes-off" approach, where the LLM does not directly ingest private examples during training, addressing a significant practical challenge in privacy-sensitive applications.

S2. DP-RFT attempts to bridge the gap between methods offering formal privacy guarantees (like DP-SGD, which requires direct data access) and methods avoiding direct data exposure but lacking domain fidelity (like Aug-PE). This is a well-identified and important problem in the field.

S3. The evaluation covers four diverse datasets (PubMed, BBC News, Wildchat, QMSum) and assesses both intrinsic quality (similarity with real data) and extrinsic utility (downstream performance). Promising results vs DP-SGD under tight privacy budgets are interesting to see. I appreciate the qualitative analysis too!

**Weaknesses:**

W1. While DP-RFT generally shows better mean/max embedding similarity, its performance on FID is comparable to or worse than Aug-PE, especially for PubMed and Wildchat. I am concerned about the overall fidelity of the synthetic data. For instance, for Wildchat (ε=∞), DP-RFT has a much higher FID (0.74) compared to Aug-PE (0.39) and even the private data (0.07).

W2. The impact of R_prompt on preventing reward hacking is not empirically demonstrated or thoroughly analyzed in the main results. The ablation study on reward function states that "the choice of reward leads to small yet not significant changes," which contradicts the stated purpose of mitigating reward hacking and raises questions about the necessity and effectiveness of this component. If the impact is small, is it truly mitigating a significant issue, or is it adding unnecessary complexity?

W3. While the ablation study shows that more powerful embedding models and language models generally lead to better DP-RFT performance, the improvement for DP-RFT is not always consistent across all privacy budgets (e.g., for gte-7B, ε=∞ is 18.51 vs ε=4 is 18.24). A deeper analysis into why Aug-PE's performance degrades with stronger models or under different privacy budgets would strengthen the discussion.

W4. While the paper emphasizes the "eyes-off" aspect, the privacy analysis relies on adding Gaussian noise to R_sim derived from nearest-neighbor votes from the private corpus. How does this process ensure that the raw private data is never "exposed" or "ingested" by the LLM during training, even implicitly through the reward signal, and how this differs fundamentally from DP-SGD's direct access. While the "Depiction Protocol" states that the image model can see the entire conversation, a visual diagram could still help to illustrate this key distinction.

W4. The paper would greatly benefit from additional figures. For instance, a diagram illustrating the flow of information for R_prompt (LLM-as-a-judge and rule-based checks) and how it integrates with R_sim to prevent reward hacking would significantly enhance understanding. Similarly, a more detailed diagram explaining the "eyes-off" mechanism and how privacy is maintained at each step of DP-RFT, especially contrasting it with DP-SGD's data flow, would be highly beneficial. The explanation of the iterative process and DP-protected nearest-neighbor votes is a bit abstract and could be clarified with a visual example.

**Questions:**

See above.

---

> ### Author Response · Authors · 2025-11-26
>
> Thank you for your careful review! We are encouraged to see that the reviewer found our paper to tackle an important problem with comprehensive quantitative and qualitative analysis.
> Please see our response below:
>
>
> [W1] Lower FiD of DP-RFT compared to Aug-PE:
> * Please refer to our general response for similarity measures of DP-RFT compared to other metrics. In summary, Aug-PE is designed to directly optimize FiD, so higher FiD on PubMed and WildChat is expected, especially since these datasets are already close to $M_{gen}$ (lower perplexity). However, FiD alone is not reliable for long-form structural text: our LLM-judge evaluation (Fig. 5) shows DP-RFT produces data more similar to the private data than Aug-PE, consistent with our downstream evaluation.
>
> [W2] The impact of $R_{prompt}$:
> * $R_{prompt}$ is proposed to prevent reward hacking on similarity w/ the embedding model with degraded quality, especially as embedding models are known to have certain bias, as we discuss in Section 3.2 of the paper. We provide an ablation study where we remove R_prompt, please refer to our general response (“Ablation on removing $R_{prompt}$”) for details.
>
> [W3] Clarification on ablation study:
> * Regarding the improvement for DP-RFT for gte-7B, we believe that the presented result shows a consistent gain for both privacy budget compared to using gte-1.5B (ϵ = ∞ 17.46 → 18.51; ϵ = 4: 17.81 → 18.24).
> * Thank you for raising the question regarding Aug-PE's performance of larger models, we will include a more detailed analysis.
>
> [W4] Clarification on the eyes-off setting:
> * Please refer to our general response  ("Clarification on our privacy setting"), and how our method ensures differential privacy.
>
> [W5] Thank you for your suggestions on additional figures, we will revise our figure 1. Please see below for detailed section of the setting presented in the current version of the paper:
> * Combining $R_{prompt}$ and $R_{sim}$: please refer to line 207-211. Specifically, we use a piecewise reward: if the prompt-similarity score $R_{prompt}$ falls below a threshold the reward is set to 0; otherwise, the reward equals $R_{sim}$​.
>
> * Thank you for the suggestions of better illustrating the privacy settings of DP-SGD and DP-RFT! The main difference is whether the private data is directly used to fine-tune the downstream language model, which we attempted to illustrate in Figure 1. We will update our figure to make that clearer.
>
> * For the process of DP voting as reward, please refer to figure 2 for the exact procedure.

---

### Official Review · Reviewer_cPfG · 2025-11-14

**Soundness:** 3
**Presentation:** 4
**Contribution:** 4
**Rating:** 8
**Confidence:** 4

**Summary:**

The authors propose to the privatized histogram signal employed in private evolution to directly RL tune the model to generate similar text to the private data.

**Strengths:**

- Experimental setups are solid, and results are convincing. Using the same generation model, embedder, and initial prompt for DP-RFT and AUG-PE, DP-RFT outperforms AUG-PE and also the baseline of just using the initial prompt with the baseline model.

- The method has numerous advantages. In addition to the fact that private data is not ever directly ingested by the model (which mitigates privacy risks from possible DP implementation gaps); the method is very simple; furthermore it can be implemented with off the shelf RL tools and hence does not require the heavy engineering investment and compute required for a DP-SGD implementation

- Ablations are good

**Weaknesses:**

- In Table 1, the private finetuning baseline has a mismatched number of samples. Although the number of samples (2k) is the same for training, AUG-PE and DP-RFT use the full dataset and associated N for noise calculation for generating the 2k, while private finetuning only gets 2k for such.

- There does seem to be a certain amount of prompt engineering and reward function crafting required to get things to work, but it is the same case in private evolution.

- No results for privately finetuning Qwen 2.5-3B to generate synthetic data in the similarity and downstream evaluation, which a very important baseline to understand the strength of the proposed method.

**Questions:**

- [naming] “Differentially private reinforcement finetuning” sounds to me like standard RL + DP-SGD. To my knowledge, no one has staked a claim on this moniker. But i believe there will be some confusion in calling the objective perturbation-based approach described in the present paper DP-RFT.
- [typo] Row 1 for Table 1 looks a bit strange. Is there a typo here?
- [typo] Indices of the summation in Line 174 look incorrect.
- Is it correct that the QWEN baseline in Table 1 should be considered epsilon=0 and should therefore be comparable to the other methods at epsilon=1? If so, this should be more accurately reflected in the table.
- Interestingly, FID seems poorer than AUG-PE despite the data capable of training better downstream next token predictors. Any possible explanations here?
- In Figure 2 and A.4, the private histogram’s sensitivity bound comes from a per-entry bound implying a norm bound. This feels loose. Why not just l2 clip the similarity vector between a private example and all synthetic example? I think this would have better noise to signal.
- How does the rollout batch size affect the results? How about private data batch size (e.g. employing privacy amplification by subsampling)?

Overall, i am very excited about this paper?

---

> ### Author Response · Authors · 2025-11-26
>
> Thank you for your review and careful feedback. We are encouraged to see that the reviewer is excited about our paper! Please see our response below:
>
> [W1] We have added the results for fine tuning on the same size of the private corpus used by DP-RFT (as reported in Table 10) in Table 1 and below. We observe that DP-RFT outperformed DP-FT when the private dataset size is small (QMSum), while DP-SGD performs better with an abundance of private training data (PubMed).
>
> | Dataset  | Method                 | **inf** | **4** | **2** | **1** |
> |----------|-------------------------|--------|-------|-------|-------|
> | **PubMed** | **DP-FT (full data)** | *44.88* | *18.41* | *17.99* | *17.63* |
> |          | DP-FT (N=2000)          | 20.97  | 9     | 8     | 6     |
> |          | Aug-PE (N=2000)         | 13.96  | 14.41 | 14.16 | 13.71 |
> |          | DP-RFT (N=2000)         | 17.46  | 17.81 | 17.32 | 16.79 |
> | **BBC**  | **DP-FT (full data)**    | *25.75* | *14.07* | *13.71* | *13.09* |
> |          | DP-FT (N=2000)          | 17.97  | 9.29  | 7.96  | 6.9   |
> |          | Aug-PE (N=2000)         | 11.63  | 10.78 | 10.03 | 10    |
> |          | DP-RFT (N=2000)         | 13.72  | 13.25 | 13.03 | 13.51 |
> | **Wildchat** | **DP-FT (full data)** | *28.78* | *13.93* | *13.52* | *12.8* |
> |          | DP-FT (N=2000)          | 19.62  | 8.28  | 7.37  | 5.3   |
> |          | Aug-PE (N=2000)         | 13.29  | 14.04 | 13.81 | 13.63 |
> |          | DP-RFT (N=2000)         | 13.93  | 14.08 | 14.14 | 13.89 |
> | **QMSum**| DP-FT (full data, N=700)|  *32.82*  | *11.52* | *9.88*  | *7.96*  |
> |          | Aug-PE (N=700)          | 8.23   | 8.53  | 7.82  | 7.63  |
> |          | DP-RFT (N=700)          | 11.03  | 10.97 | 10.95 | 11.11 |
>
>
> [W2] We acknowledge that our method depends on a reliable $R_{sim}$ that can capture the similarity of the private text. As the reviewer mentions, this is a similar assumption as prior work.
>
> [W3] We thank the reviewer for bringing up the baseline of fine-tuning QWEN-2.5-3B using DP-SGD and then training a downstream model on the generated synthetic text. Unfortunately we do not find an existing DP-SGD implementation for QWEN-2.5-3B, which is a relatively recent model. If the reviewer has a pointer for such resources, we will be more than happy to include this baseline. We would also like to highlight that the proposed baseline also exposes private text to model training, which differs from the “eyes-off” setting we are considering.
>
> [Q1] We thank the reviewer for his comment on potential confusion of our naming. When we came up with the name, we think RFT fits the spirit of this method well, i.e. In RFT (or reward-based RL), we are tuning models with only access to “reward” instead of “ground truth” examples. But yes, we see your point that it would sound like “DP-SGD”’s counterpart, we will make sure to clarify the setting and contrast with DP-SGD in the paper.
>
> [Q2, 3] Thank you for noticing this typo! R_{sim} is defined as $R_{sim} = \frac{1}{|D_{priv}|} \sum_{i=0}^{|D_{priv}|} sim(d, D_{priv, i})$, where we divide over the number of private documents. We have updated it in the paper.
>
> [Q4] Yes the reviewer is right, we have updated Table 2 in the paper to reflect this.
>
> [Q5] Please refer to our general response for discussion on FiD results compared to Aug-PE. ("FiD compared to Aug-PE")
>
> [Q6]  The reviewer is right that conducting an L2 clip of the similarity vector between a private sample and all synthetic samples could be better. We will explore it in our future work.
>
> [Q7] Thank you for your questions! We will explore the impact of rollout batch size for our training on downstream performance. Regarding private data batch size, we follow Aug-PE to use the entire private data to conduct the DP voting. While subsampling can lead to privacy benefits, it comes with loss of fidelity as the votes will be less accurate, and we are currently unsure what is the trade-off. Thank you for raising this question, we will explore this further in our future work!

---

### Author Response · Authors · 2025-11-26
**General response**

We thank the reviewers for their reviews and feedback. We are glad to see that the reviewers find that our paper address a well-motivated and important problem [6sPe, 3Y1p, XTBv, cPfG], addresses a practical challenges (“eyes-off” setting) for privacy-sensitive applications [6sPe, XTBv, cPfG] with empirical improvements over baseline methods over diverse datasets [6sPe, cPfG].

Below we provide a response to shared question raised by the reviewers.

## Clarification on our privacy setting

Reviewer 6sPe and Reviewer XTBv raise questions about our “eyes-off” premise and  concerns about privacy implications of the reward calculation of DP-RFT,  which involves DP-voting based on embedding of private examples.


**Clarification of the “eyes-off” setting**:  We would like to clarify the setting in our paper: we assume a privacy budget where the raw private example cannot never be used as input by the downstream LLM that are trained. This is a practical privacy constraint that is common in the enterprise setting  [0, 1]. Both Private Evolution [0] and AugPE were proposed for this particular kind of setting. Thus, our method and Aug-PE satisfy this requirement, while any methods require direct  access to on raw private examples (e.g. DP-SGD), or any method that requires access to logits / gradients produced by forwarding private examples through models does not satisfy the “eyes-off” constraint.

**Privacy analysis of our method**: While our method involves accessing the embedding of the private example, our method ensures differential privacy by injecting noise during the voting process, similar to Aug-PE. **Please refer to Appendix A.4 for the privacy analysis of our method.**

[0] ICLR 2024 [Differentially Private Synthetic Data via Foundation Model APIs 1: Images](https://openreview.net/pdf?id=YEhQs8POIo)

[1] [Understanding Aggregate Trends for Apple Intelligence Using Differential Privacy - Apple Machine Learning Research](https://machinelearning.apple.com/research/differential-privacy-aggregate-trends)

## Clarification of our downstream evaluation set-up:

Reviewer 3Y1p and Reviewer XTBv questions regarding our downstream evaluation set-up, suggesting to evaluate larger models and other downstream tasks. Currently we fine-tune BERT_small on the generated synthetic data and report next token accuracy on the private text set. While we agree that it will be valuable to evaluate our method on more settings, we also believe that our current set of results do demonstrate the effectiveness of our method, given that **all baseline methods are evaluated in the same setting**. We note that it is a common practice to evaluate on a relatively small scale model in the literature [0, 1, 2]. We also follow prior work on evaluating next token prediction accuracy for tasks without classification labels (e.g. PubMed). We believe these are well-established evaluation practices in the community to demonstrate the effectiveness of our method.

[0] ICML 2024 [Differentially Private Synthetic Data via Foundation Model APIs 2: Text](https://arxiv.org/abs/2403.01749)

[1] ICML 2025 [POPri: Private Federated Learning using Preference-Optimized Synthetic Data](https://arxiv.org/abs/2504.16438)

[2] NeurIPS 2023 [Training Private and Efficient Language Models with Synthetic Data from LLMs](https://openreview.net/pdf?id=FKwtKzglFb)

Below, we present new results which evaluate gpt-2 on next token accuracy, showing that DP-RFT consistently outperform Aug-PE in different privacy settings.

 | Dataset  | Method | **inf** | **4** | **2** | **1** |
|----------|--------|--------|-------|-------|-------|
| **PubMed** | DP-FT | 0.43 | - | - | - |
|          | QWEN   | 0.3304 | 0.3304 | 0.3304 | 0.3304 |
|          | Aug-PE | 0.3391 | 0.341  | 0.3357 | 0.3455 |
|          | DP-RFT | **0.3831** | **0.3737** | **0.3719** | **0.3675** |
| **BBC**  | DP-FT | 0.3913 | - | - | - |
|          | QWEN   | 0.3349 | **0.3349** | **0.3349** | 0.3349 |
|          | Aug-PE | 0.3097 | 0.3178 | 0.3154 | 0.3225 |
|          | DP-RFT | **0.3357** | 0.3281 | 0.3284 | **0.3392** |
| **Wildchat** | DP-FT | 0.51 | - | - | - |
|          | QWEN   | 0.33 | 0.33 | 0.33 | 0.33 |
|          | Aug-PE | 0.4239 | 0.4361 | 0.4365 | 0.4346 |
|          | DP-RFT | **0.4527** | **0.4542** | **0.453** | **0.4503** |
| **QMSum** | DP-FT | 0.47 | - | - | - |
|          | QWEN   | 0.24 | 0.24 | 0.24 | 0.24 |
|          | Aug-PE | 0.326 | 0.3306 | 0.3357 | **0.3402** |
|          | DP-RFT | **0.33** | **0.3368** | **'0.3433** | 0.3338 |

---

### Author Response · Authors · 2025-11-26
**General response (continued)**

## Clarification of our contribution

Reviewer 3Y1p and Reviewer XTBv raise concerns about the novelty of our method. We would like to clarify our contribution:
* We propose a new fine-tuning method for DP synthetic data generation. We gracefully disagree with the reviewers’ statement that our method is a “straightforward combination of standard PPO with a DP embedding-similarity reward”. We propose to (1) use DP-voting based on similarity with the reward as the reward and (2) add an instruction following reward to mitigate reward hacking. We also explore using similarity metrics beyond embedding similarity for structural documents such as meeting transcripts and chat logs.

* We also gracefully disagree that our method is “a minor modification of prior work”. As clarified in section 3.2, we propose two new designs compared to Aug-PE (1) using raw similarity scores as votes and (2) using DP-voting as reward for model training. We believe that the ease of implementation is an advantage of our method (as noted by Reviewer cPfG).

* We provide in-depth experiments and analysis covering various types of private data. To the best of our knowledge, we are the first to conduct privacy-preserving experiments on long-form, structural documents such as chat history and meeting transcripts.

## FiD compared to Aug-PE

Reviewer 3Y1p and Reviewer cPfG raised questions about the worse FiD we reported for PubMed and WildChat compared to Aug-PE, and how to interpret this result. Please see our response below.

* Aug-PE is designed to optimize FiD, thus it is natural that it might achieve higher FiD than DP-RFT, especially for tasks that are more familiar to the base model. As we discuss in the second paragraph of Section 5.1, we measure the distance between the target distribution and the language model ($M_{gen}$) via perplexity of the private data on $M_{gen}$ and found that Wildchat’s perplexity (4.04) is significantly lower than QMSum(10.14), with PubMed’s perplexity (6.75) lower than BBC (9.11) as well. This shows that WildChat and PubMed are easier to steer from the language model.

* As it is not easy to compare similarity of long-form structural texts, we further report reference-based LLM-judge for the generated data. Results are reported in Figure 5 in Section 1.6, showing that synthetic text generated by DP-RFT are consistently considered more similar to the private data compared to those generated by Aug-PE.


## Ablation on removing $R_{prompt}$:

Reviewer 6sPe and Reviewer XTBv raised questions about the use of R_prompt in our reward.  We conduct an ablation study which removes $R_{prompt}$ from the reward on PubMed abstract with `epsilon=inf` and report evaluation of both downstream performance and similarity measure.

While the model trained with $R_{sim}$ only achieves overall higher embedding similarity with the private corpus, the downstream performance is notably worse, this shows that solely optimizing for embedding similarity could harm performance. We would also like to highlight that our downstream experiments demonstrate that DP-RFT generates data that are more helpful than Aug-PE.

**Bert-small next token accuracy (downstream performance)**
| **PubMed** | Method                 | epsilon=inf |
|-----------|-------------------------|-------------|
|           | DP-FT                   | 20.97       |
|           | Aug-PE                  | 13.96       |
|           | DP-RFT ($R_{sim}$ only) | 16.31       |
|           | DP-RFT                  | **17.46**   |

**Embedding similarity**
| **Method**            | **Mean / max sim** | **FiD** |
|---------------------------|---------------------|---------|
| DP-RFT                    | 0.47 / 0.70         | 0.46    |
| **DP-RFT ($R_{sim}$ only)**  | **0.56 / 0.78**     | **0.45** |

---

### Meta-Review · Area_Chair_vBsy · 2025-12-12

**Summary:**

The paper proposes a method for differentially private synthetic text generation. Reviewers appreciated aspects of the work but at the same time several were critical of experimental aspects (choice of models, evaluation), the presentation of the privacy claim, and the level of novelty in the method. The paper received opposing scores from highly confident reviewers. The rebuttal was elaborate and helpful and for the most part re-clarified parts of the paper. An encompassing view of the reviews and rebuttals unfortunately leads to a reject recommendation.

**Reviewer Concerns:**

In regards to almost all reviews concerns, the authors have (helpfully) reiterated and elaborated on their original position.

**Reviewer Scores:**

The two most negative reviewers, both attesting high confidence, are unlikely to have changed their minds drastically, since their criticisms of the paper were not contradicted. One did not receive a direct rebuttal but was referred to a general response. They would have possibly risen their scores to a weak reject but that would be unlikely to change the outcome for this paper.

---

### Decision · Program_Chairs · 2026-01-26

Reject